# Last-mile delivery increases vaccine uptake in Sierra Leone

Niccolò F. Meriggi[1,2,3 ✉], Maarten Voors[2], Madison Levine[4], Vasudha Ramakrishna[5], Desmond Maada Kangbai[6], Michael Rozelle[2], Ella Tyler[2], Sellu Kallon[2,7], Junisa Nabieu[2], Sarah Cundy[8] & Ahmed Mushfiq Mobarak[9 ✉]

Less than 30% of people in Africa received a dose of the COVID-19 vaccine even 18 months after vaccine development[1]. Here, motivated by the observation that residents of remote, rural areas of Sierra Leone faced severe access difficulties[2], we conducted an intervention with last-mile delivery of doses and health professionals to the most inaccessible areas, along with community mobilization. A cluster randomized controlled trial in 150 communities showed that this intervention with mobile vaccination teams increased the immunization rate by about 26 percentage points within 48–72 h. Moreover, auxiliary populations visited our community vaccination points, which more than doubled the number of inoculations administered. The additional people vaccinated per intervention site translated to an implementation cost of US $33 per person vaccinated. Transportation to reach remote villages accounted for a large share of total intervention costs. Therefore, bundling multiple maternal and child health interventions in the same visit would further reduce costs per person treated. Current research on vaccine delivery maintains a large focus on individual behavioural issues such as hesitancy. Our study demonstrates that prioritizing mobile services to overcome access difficulties faced by remote populations in developing countries can generate increased returns in terms of uptake of health services[3].

By 10 March 2022, more than a year after COVID-19 vaccines arrived on the market, 80% of the populations living in high-income countries had received at least one dose compared with only 15% of the people in low-income countries. As of 20 November 2023, only 33% of the population in Africa had received at least the first dose of a COVID-19 vaccine[1]. Low rates of vaccination keep many countries in Africa vulnerable to the threat of disease recurrence and a renewed possibility of costly lockdowns capable of undermining employment, income generation and food security[4]. Low vaccination coverage also raises the hazard of new subvariants emerging that puts the entire world at risk[5].

To understand why vaccination rates remain low, we assembled data on vaccination beliefs, hesitancy and access from several countries in late 2021 (ref. 6). Nationally representative data from Sierra Leone revealed that obtaining access to a COVID-19 vaccine required the average person in Sierra Leone to travel three and a half hours each way to the nearest vaccination centre at a cost that exceeds 1 week of wages[2]. This finding motivated the design of an intervention we implemented in March–April 2022 in partnership with the Sierra Leone Ministry of Health and Sanitation (MoHS) and the international non-governmental organization (NGO) Concern Worldwide. The primary aim of this intervention was to take vaccine doses and nurses to administer vaccines to remote, rural communities, preceded by seeking permission and community mobilization. A cluster randomized controlled trial (RCT) across 150 communities showed that the vaccination rate in treatment villages increased by about 26 percentage points in response to this intervention. In addition, large numbers of people from neighbouring communities showed up to receive vaccines at our temporary clinics. In villages that received the intervention, the average number of people vaccinated increased from about 9 people pre-intervention to 55 people within the intervention period of about 2–3 days, at a cost of $33 per person vaccinated.

These results suggest that low vaccination rates are related to deficiencies in access and that a cost-effective intervention is capable of overcoming that deficiency. The Sierra Leone MoHS operates a network of peripheral health units (PHUs), but a significant proportion of people in Sierra Leone—particularly those in inaccessible rural areas—live outside the 5-km catchment area of any PHU. As such, interventions such as the one we conducted in communities outside PHU catchment areas are necessary to ease the burden of access.

This result carries broader implications for global public health. The child mortality rate in Sierra Leone was 10.5% in 2021 (ref. 7), as many children die from preventable diseases that immunizations and other simple interventions could address. The situation is almost as acute in neighbouring Guinea and Liberia. By contrast, efforts at community engagement in Bangladesh, including simple acts of taking maternal and child health interventions to rural populations, contributed to

[1]International Growth Centre, Freetown, Sierra Leone. [2]Wageningen University and Research, Wageningen, The Netherlands. [3]Centre for the Study of African Economies, Department of Economics, University of Oxford, Oxford, UK. [4]University of Illinois, Urbana, IL, USA. [5]Boston University, Boston, MA, USA. [6]Ministry of Health and Sanitation, Freetown, Sierra Leone. [7]University of Sierra Leone, Freetown, Sierra Leone. [8]Concern Worldwide, Freetown, Sierra Leone. [9]Yale University and Y-RISE, New Haven, CT, USA. ✉e-mail: niccolo.meriggi@economics.ox.ac.uk; ahmed.mobarak@yale.edu

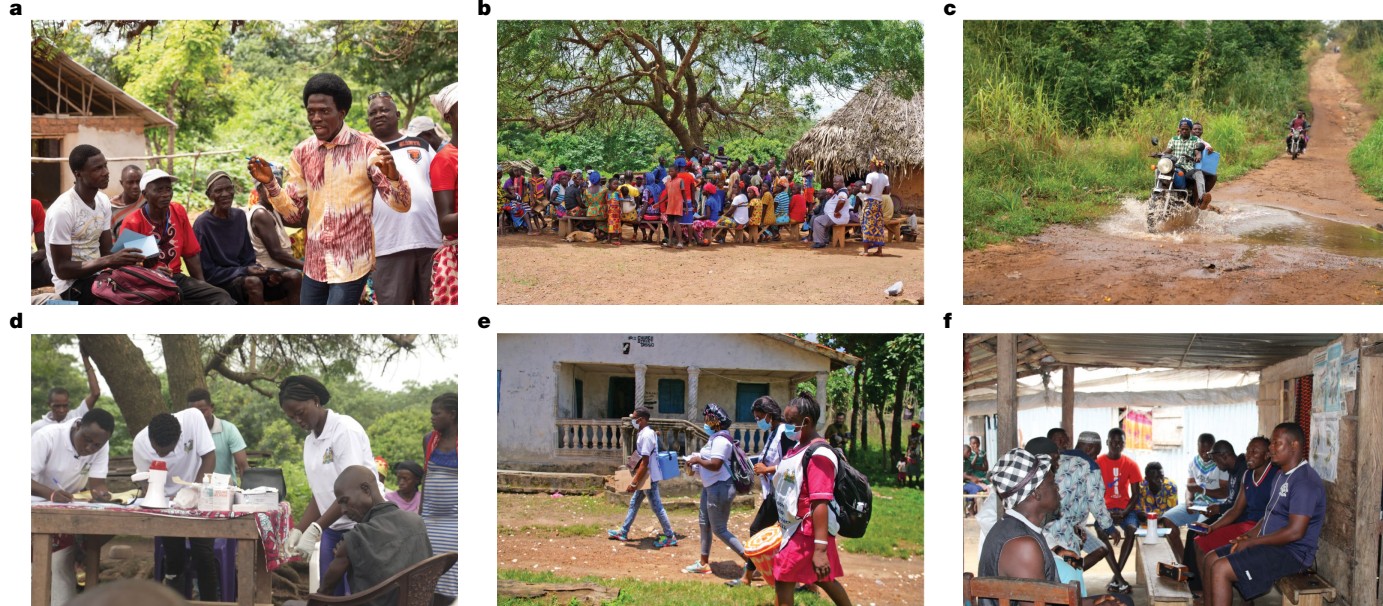

**Fig. 1 | Procedure of vaccination team visits. a–f**, Representative photographs of the steps taken by the vaccination teams in each village for the mobile vaccination clinic. **a**, Step 1. A social mobilization team from the MoHS organizes a meeting with village leaders. **b**, Step 2. Social mobilizers convene a community meeting to talk directly with all village residents about vaccine efficacy and safety, the importance of getting vaccinated, address villagers' questions and concerns, and the location and timing of the mobile vaccination site. **c**, Step 3. MoHS staff bring vaccine doses and staff to the village. **d**, Step 4. MoHS staff set up a 48–72 h mobile vaccine clinic in a central location in the village. **e**, Step 5a. Social mobilizers provide vaccine information to community members in private during door-to-door visits. **f**, Step 5b. Social mobilizers target social groups at fixed spots in and around the villages. Photographs in **a**–**f** reproduced with permission from Concern Worldwide.

increasing the infant vaccination rate from 1% in the early 1980s to more than 70% within 10 years[8]. Populations in remote areas of West Africa have proven more challenging to reach, but our intervention with COVID-19 vaccination serves as a proof of concept that it may be similarly possible to tackle the high rates of child mortality in West Africa by cost-effectively delivering simple health interventions to rural populations. In fact, bundling multiple health interventions together would further reduce the cost of delivery per person treated given the high fixed transportation costs of reaching each remote community.

These results are relevant for donors and international pharmaceutical companies who have cited cases of unused vaccines in Africa reaching expiration dates[9] to explain why low-income countries did not receive adequate supplies of vaccine doses early in the pandemic[10,11]. Our implementation efforts taught us that the Sierra Leone MoHS needed to engage in 'learning by doing' to develop new distribution systems to reach remote populations with those doses. But it is a catch-22 situation: the required experimentation is only possible once a steady and dependable supply of vaccine doses is made available.

To benchmark our results against other vaccination strategies, we conducted a comprehensive literature review that identified 235 distinct interventions in 144 RCT studies that used information, nudges, community engagement, social signalling and non-financial and financial incentives to increase vaccination rates across many settings around the world. More than one third of these interventions produced null effects. Here our access intervention produced a larger percentage point effect size than 223 (95%) of the treatments reviewed. This result is not surprising because vaccinating the first 50% of the population in remote parts of low-income countries requires solving the fundamental problem of access, which we address. Once access issues are addressed, misinformation and hesitancy may loom large in the effort to vaccinate the last 20% of the population of high-income countries who stubbornly hold out, and this is the target of the bulk of the literature. Even in high-income settings, access constraints were relevant in the earliest phase of COVID-19 vaccine delivery[12].

This finding implies that we may need to further emphasize access interventions if we are to increase the global vaccination rate and improve vaccine equity. Guidelines from the Centers for Disease Control and the World Health Organization (WHO) highlight the importance of 'bringing services closer to the people', and our RCT is a proof of concept that such approaches can increase vaccination rates rapidly and cost-effectively, even under difficult circumstances in the most remote communities. The mobile delivery concept has produced large effects in HIV testing[13]. Rigorously demonstrating effectiveness in vaccine delivery is crucial given the persistent low rates of vaccination in low-income countries. Our literature review revealed thousands of studies on vaccine hesitancy and misinformation, but only a handful on vaccine supply and access, with a clear bias in favour of high-income contexts. This imbalance is emblematic of a wider debate on the relative importance of individual-specific behavioural factors versus systemic deficiencies in limiting the diffusion of welfare-improving technologies among low-income populations[14]. Prominent behavioural scientists have recently acknowledged our excessive focus on individual behavioural peculiarities ('i-frame') at the expense of systemic solutions ('s-frame')[15].

## Context and research design

We conducted a pre-registered cluster RCT in 150 rural villages in Sierra Leone. We first mapped all PHUs where the MoHS was offering COVID-19 vaccines together with the catchment areas of a PHU, which is defined by the MoHS as the 5-mile (about 8-km) radius around each PHU. We then compiled a list of all communities situated outside these catchment areas and randomly selected 150 communities from this list. Overall, 100 communities were randomly assigned to receive the intervention and the other 50 were assigned to the control group. During March and April 2022, a research team first visited all communities to

conduct a village population listing and a baseline survey. Immediately afterwards, mobile vaccination teams coordinated by the MoHS visited the 100 villages assigned to the intervention for 2–3 days per village (Supplementary Fig. 1).

On the first day of the intervention, a social mobilization team—trained and supervised by the MoHS—organized a conversation with all village leaders, including the town chief, mammy queen, town elders, the youth leaders and religious leaders, and any other important stakeholders including the paramount and section chiefs if they were available (step 1; Fig. 1a). Members of the social mobilization team we employed were previously vetted and trained by ministry staff and commonly engaged for short-term projects such as vaccination campaigns. This cadre is referred to as MoHS volunteers because they are paid per-diem against project work and not a regular salary per a civil servant. The mobilization team explained the purpose of the visit, answered questions about the available vaccines and asked leaders for their cooperation in encouraging eligible community members to take the COVID-19 vaccine.

Social mobilizers then asked leaders to convene a community meeting that same evening (when people return home from farms) to allow mobilizers to talk directly with all village residents about vaccine efficacy and safety, the importance of getting vaccinated, and to address villagers' questions and concerns. This process (step 2; Fig. 1b) ended with social mobilizers explaining the location and timing of the mobile vaccination site that they were about to set up.

Vaccine doses, nurses to administer vaccines and MoHS staff who could register the vaccinated were brought into the community either the same evening or early the next morning (step 3; Fig. 1c). The vaccine doses and staff often travelled on motorbikes or on boats given the difficult terrain they had to traverse to reach these remote communities. Once the team was in place, the temporary vaccination site started operating in a central location in the village (step 4; Fig. 1d). Villages in our sample were small, with houses closely clustered; therefore, walking distances to the vaccination site were short. The vaccination site remained operational from sunrise to sunset over the next 2–3 days, which enabled people to visit when convenient. Nurses and registration staff remained stationed at the temporary clinic while the mobilizers continued to provide vaccine information to various community members (step 5).

We randomized the exact nature of these additional step-5 mobilization activities. Half the treatment villages were randomized into an individualized door-to-door campaign (step 5a; Fig. 1e), whereby social mobilizers went to 20 randomly selected structures to privately discuss any concerns about that vaccine that the household residents had and to encourage them to visit the vaccination site. The other 50 treatment communities were randomized into small-group outreach (step 5b; Fig. 1f), whereby mobilizers targeted social groups who gathered at fixed spots in and around the villages (for example, groups of farmers in fields, mosque attendees or women collecting water). Social mobilizers engaged the group to have joint conversations about the vaccines. There was equipoise about whether individualized or small-group outreach would be more successful in persuading people to get vaccinated, so we tested both strategies.

## Effects on COVID-19 vaccination rate

Our primary outcome was verified adult vaccine uptake, which was measured using a respondent-level question on whether the person took a COVID-19 vaccine of any type, checked against their vaccination card (if consented). This measure provided us with a site-level count of vaccine doses administered.

To calculate a village-level vaccination rate, we first enumerated the population in all 150 treatment and control villages. Such community census lists typically do not exist in Sierra Leone. Our research team therefore walked to all structures in every village to tally the number

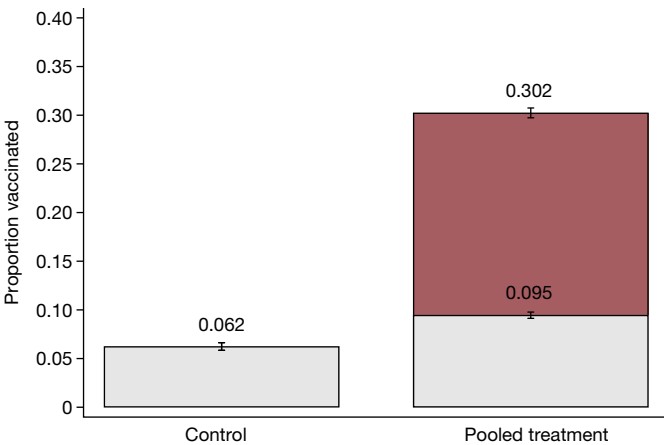

**Fig. 2 | Vaccination rate among adults enumerated during the census before and after mobile vaccination programme.** The proportion of vaccinated adults that were enumerated during the census before and at the end of the study in control villages and pooled treatment villages. The analysis includes the 12,096 people (aged >18 years) in 150 villages. Data are presented as the mean ± s.e.m. In the control group, 6% were vaccinated at baseline, whereas 9.5% were vaccinated in treatment groups. At endline, 30% were vaccinated. The intent-to-treat treatment effects estimated using OLS and including randomization block fixed effects and heteroscedasticity-robust standard errors clustered at the village level are provided in Extended Data Table 3.

of households (39 on average, s.d. = 23), and the number of individuals living in those households (29,588 individuals across the 150 villages, or about 197 people per village).

The population of these villages was on average 22.3 years old, 26.5% of households were headed by women and 64.5% of people lived in a household of 6 or fewer people. Only 20.1% lived in a household where the household head had any form of formal schooling, and about 86.1% lived in a household where the head was primarily engaged in farming. Respondent characteristics were well balanced across the treatment groups (Extended Data Table 2) except for the following: the baseline vaccination rate; the proportion of households employed in agriculture; the proportion of households that own a radio; the proportion of women breastfeeding; and the proportion that owns land. Although an overall $F$-test did not reject the equality of means across the full set of outcomes, we added these covariates in part of our analysis below.

Figure 2 shows that at baseline, the average vaccination rate in villages assigned to the control group (control villages) was 6.2% compared with 9.5% in villages that received treatment (treatment villages) (ordinary least squares (OLS) regression, difference = 0.035, s.e. = 0.014, $P = 0.015$, $n = 12,096$). After intervention, the vaccination rate increased to 30.2% in treatment villages. We report effects from linear regression specifications of the intent-to-treat effect with randomization fixed effects and heteroscedasticity-robust standard errors clustered at the village level in Extended Data Table 3 (see the section 'Statistical analysis' in the Methods). The intent-to-treat effect was 26 percentage points (OLS regression, s.e. = 0.018, $P < 0.001$, $n = 12,096$). The results remained qualitatively similar (OLS regression, 25 percentage points, s.e. = 0.019, $P < 0.001$, $n = 12,096$) when covariates for respondent characteristics were added that were imbalanced at baseline (such as vaccination status) or when we aggregated the data up to the village level (OLS regression, 28 percentage points, s.e. = 0.025, $P < 0.001$, $n = 150$).

This increase in the vaccination rate is an underestimate of the total number of vaccines administered over those 2–3 days, as it does not include vaccines given to migrant returnees and others from nearby villages. The average uptake also masks considerable heterogeneity among villages. In 2 out of the 100 treatment villages, there was no increase in vaccination rate because the village authorities either dissuaded villagers from getting vaccinated or refused permission for the

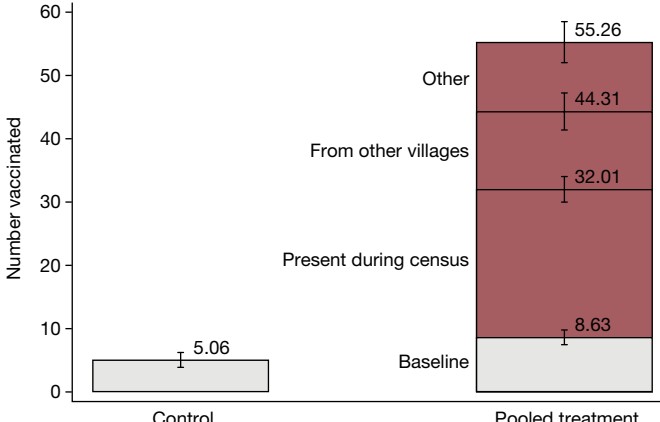

**Fig. 3 | Count of people vaccinated per site before and after the mobile vaccination programme.** The number of the people vaccinated before and by the end of the study. Data are presented as the mean ± s.e.m. The analysis includes 150 villages. In the control group, on average five people were vaccinated, whereas in treatment villages, this was nine people. Treatment increased the count to 55 people, including 22–23 individuals who were enumerated during the census group, 12–13 people from nearby villages and 11–12 short-term, circular commuters or migrant returnees who were not present on the day of the census and could not be matched to our listing records, as well as individuals whose community of origin was unknown. The intent-to-treat treatment effects estimated using OLS and including randomization block fixed effects and heteroscedasticity-robust standard errors are provided in Extended Data Table 4.

intervention to take place, which causing the intervention to essentially fail at step 1 (Fig. 1a). By contrast, the full distribution of vaccination rates displayed in Supplementary Fig. 2 shows that in 5 villages, more than 50% of adults enumerated in the community census were vaccinated during the course of our intervention. A similar large degree of variation was evident from the total count of immunizations set per village (Supplementary Fig. 3).

## Effects on total vaccination count

Many of the people who attended our temporary clinics to receive a vaccine were not enumerated during the community census. These additional people fell into one of three categories: residents of other nearby villages (who heard about the clinic and were interested in taking advantage of the easy access to a vaccine); recent migrant returnees who were not present during the village listing; and others—such as high-frequency commuters—not captured in the census. For these auxiliary populations, we do not have a denominator and can therefore not estimate a vaccination rate. We can, however, provide results on vaccination counts.

At baseline, there were on average about 5 people vaccinated in control villages and about 9 people in treatment villages (OLS regression, difference = 3.57, s.e. = 1.51, $P < 0.021$, $n = 150$). Figure 3 shows that after the intervention was implemented over the subsequent 2–3 days, the number of vaccinated individuals increased to about 55 people on average per treatment site, which is a 6-fold increase. This is the full effect of our mobile vaccination drive. Among individuals vaccinated who were not enumerated in the census, 53% (12–13 people per treatment community) were visitors who came from nearby villages to get vaccinated, whereas the remaining 47% (11–12 people) included short-term, circular commuters or migrant returnees who were not present on the day of the census and could not be matched to our listing records, as well as individuals whose 'community of origin' was unknown. The intent-to-treat regression estimates with heteroscedasticity-robust standard errors and additional covariates are included in Extended

Data Table 4. In total, the teams vaccinated 4,771 people aged 12 years or above. Of these, 39% received a Johnson & Johnson vaccine, 29% Pfizer, 17% Sinopharm and 16% received AstraZeneca. A variety of vaccine types was administered because there was no steady supply of any specific type of vaccine dose in Sierra Leone when this intervention was conducted. Therefore, we had to make use of the vaccines available in the Ministry of Health stocks in any given week.

## Effects of home visits

Both types of mobilization activities implemented in step 5 had similar effects on the vaccination rate. The evidence on whether the door-to-door or small-group activities were more effective was mixed. When we compared across communities, the door-to-door programme increased the adult vaccination rate by about 29 percentage points compared with 23 percentage points in villages assigned to the small-group mobilization activities ($t$-test, difference = 6 percentage points, $P = 0.014$, $n = 12,096$; see column 1 in Extended Data Table 1). However, when we studied individual households randomly assigned to a visit against those who are not within door-to-door villages, we did not detect any differential uptake. In these 50 villages, up to 20 randomly selected structures were visited for a private or semi-private conversation with residents about the vaccine and to encourage them to visit the temporary clinic. The random selection of structures enabled us to report experimental results on the effects of receiving this extra nudge on the propensity to receive a vaccine. We interpret this activity as a 'demand-side treatment', in that the visit and conversation provides that resident an opportunity to discuss their concerns or questions about vaccines in private, which could be useful to overcome potential hesitancy. Column 3 in Extended Data Table 1 shows that this extra effort did not generate additional demand beyond the effect of our 'supply side' activities to enhance vaccine access. The adult vaccination rate at the end of the vaccination programme among those who received the home visit by mobilizers was not different from those who did not receive the extra nudge points (OLS regression, difference = −0.01, s.e. = 0.019, $P = 0.543$, $n = 3,760$). Social mobilizers received extensive training and close supervision, but the lack of impact from this additional demand-generating activity on vaccine uptake may reflect low effort by social mobilizers. Within-village spillovers may also dampen these individual treatment effects. Unfortunately, we lack data on distances and other channels of interactions among households to formally test this measure. However, this type of spillover may be small owing to the relatively short time interval between the home visits and the vaccine drive.

We do not have an equivalent analysis of the individual effect of the small-group treatment because that was not randomized within villages. Moreover, the enumerators were not able to exactly track which households participated in the small-group sessions.

## Mechanisms

Although our vaccine access intervention significantly raised the vaccination rate, it was also clear that we remained far short of reaching the WHO goal of near-universal uptake. We collected individual-level data in all treatment villages after the intervention from both vaccine takers and non-takers. These data can shed some light on why and how our access intervention was more or less successful for certain types of people.

### Meeting attendance

Step 2 of our intervention (Fig. 1b) was to organize a community-wide meeting to inform all village residents about the vaccine clinic. The field team registered which community members attended that meeting, and 41% of households participated in these meetings. Overall, 44% of those who chose to attend the meeting subsequently chose to

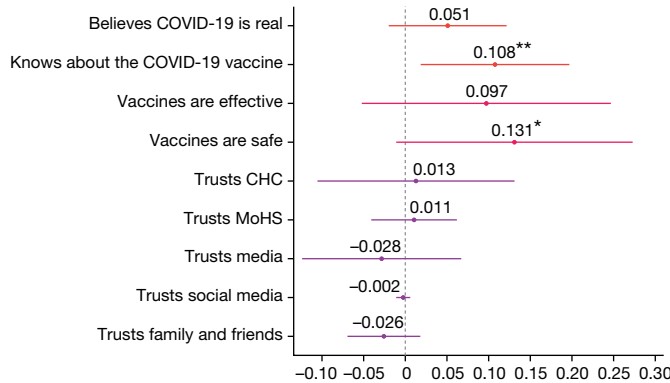

**Fig. 4 | Effect of pooled treatment on knowledge and attitudes among adults enumerated during census.** Intent-to-treat estimates of community treatment assignment for each outcome listed on the *y* axis. Treatment effects were estimated using OLS and included randomization block fixed effects and heteroscedasticity-robust standard errors clustered at the village level. Each dot is labelled with the exact coefficient (to three decimal places) and significance at the ***1,**5 and *10 per cent critical level. Bars represent 95% CIs of treatment estimates. The analysis includes 45 villages and 817 households surveyed at endline for which we observed complete randomization blocks. Associated regression results are provided in Extended Data Tables 6 and 7, including corresponding sample sizes. Reported estimates do not correct for multiple hypothesis testing. The Extended Data tables report the associated FDR-adjusted *q* values. The survey measures for the 'Believes COVID-19 is real' comes from the survey question: "Do you believe that COVID-19 exists in the world?" (yes or no). 'Knows about the COVID-19 vaccine' comes from the survey question: "Do you know about the COVID-19 vaccine/marklate?" (yes or no). 'Vaccines are effective' is 1 if respondents completely agree with the statement: "Vaccines are effective." 'Vaccines are safe' is 1 if respondents completely agree with the statement "How much do you agree with this statement: vaccines are safe." Trust in sources of information are from a multiple-select question: "Who do you most trust getting information about COVID-19?" (community health clinic (CHC), MoHS, media (news, TV), social media (Facebook, among others), family and friends, among others).

get vaccinated. One cannot impose any causal interpretation to this correlation because people who were already interested in getting vaccinated may have been the ones who chose to attend the meeting.

We can make a slightly stronger inference by examining the subset of people who stated in our baseline survey that they were unwilling to receive a vaccine (Extended Data Table 5). Within this subgroup, 53.8% of those who attended meetings ultimately took the vaccine, whereas the vaccination rate was only 14.4% among those who did not attend. Even within the converse subgroup (those who stated at baseline that they were willing to take the vaccine), meeting attendance was strongly predictive of subsequent vaccine uptake: 64.6% vaccination rate among attendees and 39.4% among non-attendees.

These are not causal estimates, but the size and direction of these correlations suggest that the information shared in the meeting, and the answers that were provided to the community's questions, are unlikely to have dissuaded people from getting vaccinated. These correlations—combined with our team's on-field experience—suggest that holding these meetings was helpful and form a necessary part of any access intervention. Encouraging greater attendance in meetings in any future replications would probably be a good idea.

### Vaccination knowledge and trust

We also collected data on another intermediate outcome in a subset of villages: people's knowledge and attitudes regarding the COVID-19 vaccine. Figure 4 shows that the treatment improved people's knowledge about vaccines (OLS regression, difference = 0.11 points, s.e. = 0.044, *P* = 0.019, *n* = 817). The change in knowledge implies that our intervention was not solely about improving access. That

is, the community interactions and the information we shared were also relevant parts of the intervention package. However, there was no significant change in people's beliefs about vaccine efficacy (OLS regression, difference = 0.097, s.e. = 0.074, *P* = 0.197, *n* = 686). Using a 95% confidence interval (CI), we can reject that our treatment increased beliefs about vaccine efficacy by more than 12 percentage points. The effect on beliefs about the safety of vaccines is not statistically precise (OLS regression, difference = 0.131 points, s.e. = 0.070, *P* = 0.069, *n* = 686)—we can neither rule out a null effect nor a 27 percentage point effect. The null effects in OLS regressions on the sources that people trust the most for receiving health information were more precisely estimated: community health clinics (OLS regression, difference = 0.013, s.e. = 0.059, *P* = 0.828, *n* = 817, 95% CI upper bound = 0.13); the MoHS (difference = 0.011, s.e. = 0.025, *P* = 0.682, *n* = 817, 95% CI upper bound = 0.06); media (OLS regression, difference = −0.028, s.e. = 0.047, *P* = 0.553, *n* = 817, 95% CI upper bound = 0.07); social media (OLS regression, difference = −0.002, s.e. = 0.004, *P* = 0.555, *n* = 817, 95% CI upper bound = 0.006); or family and friends (OLS regression, difference = −0.026, s.e. = 0.022, *P* = 0.242, *n* = 817, 95% CI upper bound = 0.018). Extended Data Tables 6 and 7 provide the associated regression estimates. Note that because this is an exploratory exercise in which we test treatment effects across several outcomes, the tables report the false discovery rate (FDR)-adjusted *q* values to adjust for multiple hypothesis testing.

### Heterogeneity across demographic groups

Figure 5 shows the differences in treatment effect for specific demographic subgroups. Extended Data Table 8 and Supplementary Table 1 provide associated regression results. The treatment effect was 7 percentage points larger for men than for women (OLS regression, difference = −0.067, s.e. = 0.016, *P* < 0.001, *n* = 12,096), and 12 percentage points larger for the >55 years age group compared with the 18–24 years age group (OLS regression, difference = −0.122, s.e. = 0.028, *P* < 0.001, *n* = 12,096). We cannot reject the null hypothesis that there is no difference in treatment effects across education (OLS regression, difference = −0.003, s.e. = 0.018, *P* = 0.864, *n* = 12,096, 95% CI upper bound = 0.032), land ownership (OLS regression, difference = 0.038, s.e. = 0.035, *P* = 0.268, *n* = 2,674, 95% CI upper bound = 0.11) or food security status (OLS regression, difference = −0.006, s.e. = 0.032, *P* = 0.865, *n* = 2,674, 95% CI upper bound = 0.06), and can rule out large effect sizes.

### Comparison with other vaccination efforts

As shown in Extended Data Table 3, the intervention increased vaccination rates by about 26 percentage points. Although such an increase seems substantial, this is the gain off a very low base rate: just 6–9.5% were vaccinated at baseline. Another relevant benchmark is our percentage point effect size and how that compares with other vaccination campaigns evaluated in the literature.

We conducted a literature review of vaccination strategies that have been evaluated using RCTs (see the section 'Literature review of vaccination uptake RCT studies' in the Methods). The Methods provides inclusion criteria for this review. We identified 144 different published RCT studies that report the results of 235 distinct interventions.

These interventions varied across multiple dimensions, spanning time, space and strategy, often as part of the same study with multiple components. For clarity and brevity, we identified five major intervention 'families', which could be further fragmented into more granular intervention 'types'. The families into which interventions were sorted were education, community actions, communications, incentives, and healthcare improvement and worker training. Among the 144 relevant studies, only 3 focused on the essential theme of vaccine access in a low-income context, and none of them were centred on COVID-19 vaccines[16-18].

Supplementary Fig. 5 demonstrates the heterogeneity of effect sizes across these 235 different treatments (effect size range of −6 to 50 percentage points). Of all treatments reviewed, 35% had no significant effect on vaccine uptake. Perhaps unsurprisingly, owing to the variety of incentive types and sizes, the incentives group was strongly positively skewed, accounting for five out of the top ten effect sizes overall. However, the highest median effect size was among educational interventions (median = 6.25)[19,20].

The intervention we conducted in Sierra Leone—whereby mobile health teams visited remote communities for 48–72 h to ease access burden—produced a larger percentage point effect size than 223 (95%) of the treatments reviewed.

Supplementary Table 2 provides details of the intervention approach used in each study. The majority of these studies were conducted in high-income settings (83%). Many of the vaccination campaigns evaluated were nudges and reminders by text messages, telephone or mailings (50%). Nudges are inexpensive, but often produce small or null effects. Other strategies involved visiting parents to educate them about the benefits of childhood immunization (25%) or sending community health workers (5%). Others offered direct financial incentives against a verified vaccination (9%).

Of special interest were recent studies that attempted to promote COVID-19 vaccinations. A study in Sweden[21] offered monetary rewards of $24 to receive a COVID-19 vaccine, and this increased the vaccination rate by an extra 4 percentage points (from 72 to 76%). A financial incentive of $10–50 combined with other nudges in the United States did not produce any effect[22]. City-wide and state-wide lotteries offering financial rewards in the United States[23,24] produced small or negative effects. Text-based reminders in the United States[25,26] and defaulting people into a vaccination appointment in Italy (so that they are forced to opt out)[27] increased vaccination rates between 0 and 3.5 percentage points.

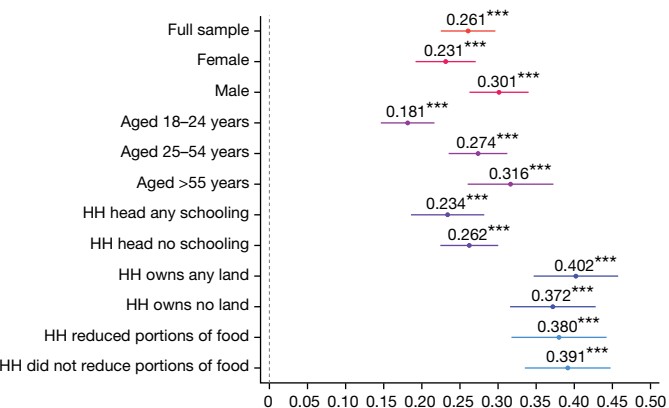

**Fig. 5 | Effect of pooled treatment by respondent characteristics among those enumerated during census.** Intent-to-treat estimates of the vaccination rate of the pooled treatment groups for each subgroup listed on the y axis. Treatment effects were estimated using OLS and included randomization block fixed effects and heteroscedasticity-robust standard errors clustered at the village level. Each dot is labelled with the exact coefficient (to three decimal places) and significance at the ***1, **5 and *10 per cent critical level. Bars represent 95% CIs of treatment estimates. The dependent variable is the vaccination status of adults at the end of the study enumerated during the census. Sex, age and schooling data are from the census. Land ownership and food insecurity are from the baseline sample. Associated treatment estimates and associated sample size for each subgroup are provided in Extended Data Table 8. The indicator for 'HH head any schooling' indicates whether the household (HH) head had schooling above the primary level. The 'HH owns any land' indicates whether the household owns land. The 'reduced portions of food' statements indicate whether any household member had reduced food portions during the previous week.

## Cost-effectiveness relative to other strategies

Sending text message reminders or running city-wide lotteries are relatively inexpensive to implement, whereas delivering vaccines to remote areas is costly. It is therefore useful to compare not just percentage point effect sizes but also the cost of administering various programmes per vaccinated individual. Moreover, we chose to work in the most remote areas not covered by the Sierra Leone MoHS vaccination programmes precisely because they are too far away even from PHUs. We collected detailed cost data on our programme to compute this metric and compared it to other studies that provide such cost information (see the section 'Literature review of vaccination uptake RCT studies' in the Methods).

The total costs of our intervention to reach 100 villages was $156,023.5, or approximately $1,560 per village. This included all travel, administration and management and supervision costs, but excluded the cost of the vaccine doses, which were provided to Sierra Leone by the COVAX programme for free. This translates to a cost per dose administered of about $33.

Extended Data Table 9 provides a detailed breakdown of the fixed and variable components of our implementation costs. Of the $33, around 27% ($9) was fixed costs of training project staff and 73% ($23) was variable costs. The most expensive category (38% or $12.50) was transportation to these remote villages, which included the cost of renting vehicles and fuel. Salaries and subsistence allowances for the social mobilization and vaccination teams accounted for another one quarter of the total costs.

To conduct this intervention again at larger scale, the variable costs would need to be repeated, but not the fixed costs of training. At scale, the cost of this intervention would therefore approach about $23 per person vaccinated. The wide availability of a cadre of staff known as Ministry of Health volunteers—individuals already vetted by the ministry and available to work as mobilizers on special projects against

per-diems—increases the potential for scaling this project nationwide in Sierra Leone. One potential challenge of replicating this project to other countries is to find trained staff who can take on that mobilization role.

Note that here we are looking at cost-effectiveness from the perspective of the planner (that is, the government) and do not consider the costs imposed on households. Depending on context, meeting attendance can be inconvenient or costly. In our context, villages are small. On average, people had to walk less than a couple of hundred metres to attend the meetings. Also, to minimize the inconvenience, meetings were held in the early evenings after people returned from their farms. As a result, the opportunity cost of time was low for most participants of the meeting.

Figure 6 provides the cost per vaccinated person in year 2000 US dollars for the subset of studies in Supplementary Table 2 that reported sufficiently detailed cost information for us to be able to compute this metric. Of the 235 different treatments identified in our literature review, only 33 (14%) directly stated the cost of the intervention per successfully administered vaccination. Furthermore, of these 33 interventions, 7 did not report a cost specific to the treatment group, but only the overall cost averaged over all groups of the study. In total, 57% of the vaccination campaigns exceeded our $33 benchmark. The mean value in Fig. 6 is $83 (s.d. = 132), even after excluding the most expensive approach.

A study in rural India[28] pursued a similar strategy to ours by setting up measles vaccination clinics. That treatment cost $75 (in 2022 dollars) per vaccine administered, but adding an incentive for the parents to bring their children to the clinic lowered the cost to $38 per child vaccinated. The only other COVID-19 vaccine study in our review to provide cost information[21] offered $24 as a financial incentive to get vaccinated in Sweden. Unfortunately, that study did not report costs of other components of the programme, such as the cost of administering the incentive programme, verifying individual-specific vaccination

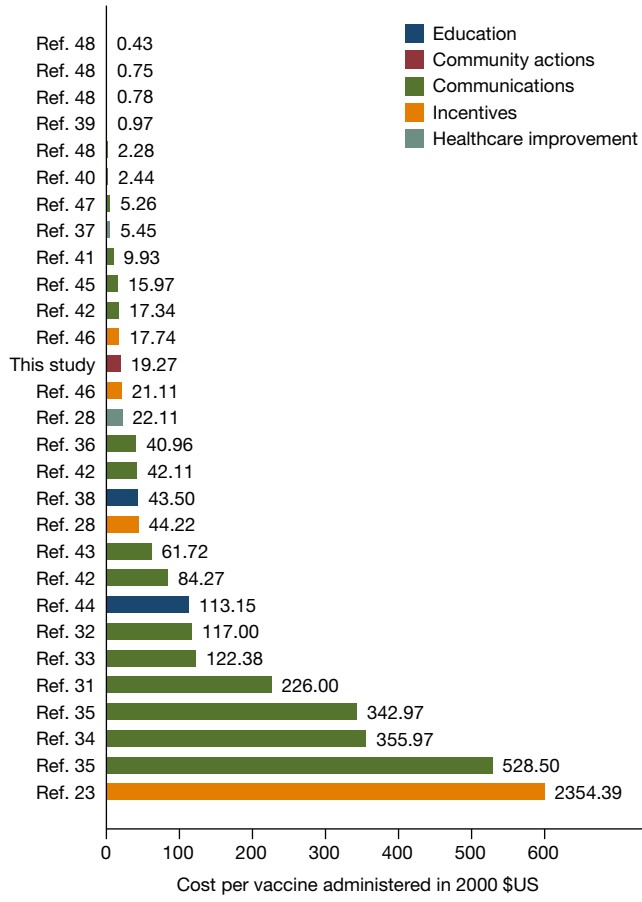

| | |
|---|---|
| Ref. 48 | 0.43 |
| Ref. 48 | 0.75 |
| Ref. 48 | 0.78 |
| Ref. 39 | 0.97 |
| Ref. 48 | 2.28 |
| Ref. 40 | 2.44 |
| Ref. 47 | 5.26 |
| Ref. 37 | 5.45 |
| Ref. 41 | 9.93 |
| Ref. 45 | 15.97 |
| Ref. 42 | 17.34 |
| Ref. 46 | 17.74 |
| This study | 19.27 |
| Ref. 46 | 21.11 |
| Ref. 28 | 22.11 |
| Ref. 36 | 40.96 |
| Ref. 42 | 42.11 |
| Ref. 38 | 43.50 |
| Ref. 28 | 44.22 |
| Ref. 43 | 61.72 |
| Ref. 42 | 84.27 |
| Ref. 44 | 113.15 |
| Ref. 32 | 117.00 |
| Ref. 33 | 122.38 |
| Ref. 31 | 226.00 |
| Ref. 35 | 342.97 |
| Ref. 34 | 355.97 |
| Ref. 35 | 528.50 |
| Ref. 23 | 2354.39 |

Legend:
- Education
- Community actions
- Communications
- Incentives
- Healthcare improvement

Cost per vaccine administered in 2000 $US

**Fig. 6 | Cost per person vaccinated compared with other studies.** The cost per vaccination administered (in year 2000 US dollars, calculated using inflation data sourced from the US Bureau of Labor Statistics). These are treatments from studies included in Supplementary Table 2 that explicitly provided information about the cost of the intervention per vaccine actually administered. This cost specifically refers to the intervention and does not include the cost of the vaccine itself. Studies that did not unequivocally state the cost of the intervention per vaccinated person were not included. The colour of each bar indicates the broad type of intervention. The cost per person vaccinated in our study was $32.70, which is approximately $19.27 in 2000 US dollars. The mean value in this figure is $83 (s.d. = 132) after excluding the most expensive approach. References 23,28,31–48 are cited in this figure.

information in the administrative records, sending two text message reminders, among others.

## Discussion

Our findings showed that a simple last-mile vaccine intervention tripled vaccination rates within 48–72 h. In addition, auxiliary populations showed up to take advantage of these mobile vaccination centres, which more than doubled vaccination counts. Our intervention, despite being delivered in highly remote locations, is cost-effective relative to other efforts that aim to increase vaccination rates.

### Policy implications

Vaccine equity remains an important policy goal[10]. Vaccination rates are severely lagging among rural populations in Africa. Therefore, achieving equity requires us to devise an effective strategy to reach this population. Our study provides some guidance on how to formulate that strategy.

The most immediate and direct implication of our results is for the government of Sierra Leone to replicate and expand this cost-effective programme to reach the 59% of the country's population who reside in similar remote, rural areas outside PHU coverage. The largest expense of our intervention was the transport cost of reaching remote communities; therefore, an obvious implication is that we should bundle COVID-19 vaccines with other necessary mother, infant and child health interventions that can be simultaneously delivered on the same trip[3]. Such an approach could substantially reduce costs per person treated. However, this would still be expensive for a resource-constrained MoHS to launch at scale, and international partners must provide support. A recent study in rural Western Kenya[29] demonstrates that such integrated approaches, combining HIV testing with other preventive health services such as bed nets and water filters, can be successfully implemented.

We have begun building the necessary coalition to implement such a bundling strategy to improve the cost-effectiveness and scalability of this last-mile-delivery intervention. The Sierra Leone MoHS has prioritized HPV vaccination for girls aged 10–12 years, and routine immunizations (DTP, measles, polio) for children aged 0–6 years to bundle with any further COVID-19 vaccine delivery. It is reasonable to wonder whether COVID-19 vaccine distribution is a high-priority investment given the low incidence of COVID-19 in Africa. However, as the experience in India from April 2021 shows, new COVID-19 variants have the capability to devastate public health systems in developing countries[30]. Health infrastructure in a typical country in Africa is even more fragile than it is in India. If we pay the transport cost to take a bundle of health interventions to these remote communities, COVID-19 vaccines could easily be an element of that bundle.

The other direct implication is to replicate such a programme in neighbouring countries with similar last-mile delivery challenges. The majority of people in sub-Saharan Africa live in rural areas[7], so overcoming access challenges through such initiatives holds enormous potential for both achieving vaccine equity and maximizing global coverage.

Our study showed that low-income countries need to experiment with creative ideas to overcome logistical challenges, such as setting up temporary clinics and sending both vaccine doses and nurses to remote locations on motorcycles. A broader implication for international development partners and pharmaceutical companies is that they need to facilitate and underwrite such experimentation by making vaccine doses and budgets readily available to allow ministries of health to learn what approaches work best in a given context. Local institutions need to engage in 'learning by doing', which is impossible without a reliable supply of vaccines and incentives for staff to tinker with new, innovative ideas.

### Study limitations

The intervention we implemented had two important limitations. The $33 cost (per person vaccinated) varied substantially across villages because the number of individuals per village that we managed to vaccinate varied (the vaccination rate ranged from 0 to 69%). Village leaders did not allow us to conduct the intervention at all in 2 of the 100 treatment villages, which inflates the overall average cost of our intervention. Any replication should try to identify early the villages where such refusals might occur and find ways to avoid having the entire vaccination team travel there.

Second, we observed large cross-team variation in performance. Supplementary Fig. 4 shows that some of our teams administered more than twice as many vaccines as other teams (the number of vaccines administered ranged from 0 to 146 per village, mean = 48, s.d. = 31, median = 42). Some of these differences could be due to differences in village characteristics, but our implementation experience suggests that variability in team effort also played a part. Providing good performance incentives to teams could improve the cost-effectiveness of this strategy. Given that a large proportion of the expense of the intervention is the cost of travelling to the remote village, we should

strategize to ensure that we maximize the vaccination rate within the 48–72 h window once we get there.

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

## Methods

### Ethics approval

We received Institutional Review Board (IRB) approval from the Sierra Leone Ethics and Scientific Review Committee (SLERC 20220210), Yale University (2000031541) and Wageningen University (WUR 20222222). The research protocol was pre-registered at the ISRCTN Registry (study identifier ISRCTN17878735). All study participants completed informed consent.

The study was implemented in close collaboration between the researchers, the Government of Sierra Leone's Expanded Programme on Immunization (EPI) at the MoHS, their National COVID-19 Emergency Response Centre and Concern Worldwide (an international NGO who partners with MoHS on health projects). This collaboration came together because all partners had the joint goal of addressing barriers to vaccine adoption in rural Sierra Leone. Although all partners are responsible for the research design, only the Ministry of Health team was responsible for actually distributing and administering vaccines. We had a memorandum of understanding in place to govern this collaboration.

### Village study sample

To determine the sample size, we ran a power calculation assuming a 5% significance level with 80% power. We assumed an intra-cluster correlation of 0.15 as decisions to take a vaccine are probably highly correlated within a village. Average village populations are 2,480 people. We assumed an eligible population of 50% and a baseline vaccination rate of 2.5%. Based on the treatment effects reported in the literature for similar studies, we took a conservative approach and set our expected minimum detectable effect at 0.05. We oversampled slightly and the final design included 150 communities across the three treatment groups in a 1:1:1 ratio.

We chose study sites in collaboration with the MoHS. We started with the 2015 Sierra Leone census, which contains data on 20,659 communities in 166 chiefdoms across 16 districts. We selected 7 largely rural districts (Koinadugu, Falaba, Karene, Kambia, Tonkolili, Bombali and Port Loko), limiting the sample to 8,784 communities in 54 chiefdoms. We then restricted our sampling frame to communities that, according to the 2015 census, had no health clinic within 5 miles (about 8 km) of the community centre, the standard PHU catchment area (Extended Data Table 10), resulting in 1,849 communities. From this list, we excluded very small communities that contained fewer than 19 structures and communities for which latitude and longitude were missing. The final sampling frame consisted of 420 communities located in 49 chiefdoms and 7 districts. Within each district, we then matched communities on the following strata: (1) the share of the population that was immunized; (2) the age of the population; (3) literacy levels; and (4) the distance from the closest clinic. This allowed us to identify communities that had the most similar characteristics within a district and used this to assign the most similar communities to one of the treatment groups and establish comparable 'triplets'. This resulted in 106 triplets in total. We then randomly selected 50 triplets using district as a blocking variable. The final list included 9 triplets each for Koinadugu and Falaba districts, 8 triplets for Karene district and 6 triplets each for Port Loko, Tonkolili, Kambia and Bombali districts.

### Randomization

**Randomization to vaccine access treatments.** Within each of the 50 triplets, we randomized villages into control, door-to-door and small-group treatment groups. This resulted in 50 villages assigned to control, 50 to door-to-door and 50 to small group (Supplementary Fig. 1). The sample was well balanced on observable characteristics (the $F$-statistic at the bottom of Extended Data Table 2 is small and not significant).

**Household-level random assignment to door-to-door treatment.** Within the villages randomly assigned to the door-to-door treatment group, we randomly selected up to 20 residential structures from the community census list to receive a visit from the social mobilization team.

### Data collection

**Community census listing and baseline survey.** Before any intervention activities took place, the research team implemented a community census to enumerate all households in all 150 villages. The research team went door to door to each residential structure and asked how many households resided in the structure. They then interviewed each household head to create a roster of those who 'eat from the same pot; and reside under the same roof for at least the past 9 months (aside from newborn babies).' For each household member, enumerators asked about the sex, age and vaccination status. The total census includes $N = 29,588$ people. Migrant household members who were temporarily away on the day of the visit would have been missed from this listing.

Next, the research team randomly selected a sample of 20 households per village from the households listed in the census to conduct a short (baseline) survey with the household head to record household characteristics (age, sex and education), access to land and food security. The total baseline sample included $N = 2,240$ respondents.

**Exit and endline surveys.** After the interventions were implemented, the research team conducted an exit survey of those who received a vaccine at each mobile vaccination clinic. The survey recorded the vaccination status verified using visual inspection of the vaccination card, as well as age and sex.

During the exit survey, enumerators also recorded where people came from and their district and village name (if different from the implementation site). To assess between-village spillovers, we then matched the names of reported villages back to our list of control villages. Using a hard match on district names and then a Levenshtein distance metric to match village names, allowing for a string distance of 2, we found only 8 matches. Using a more conservative cut-off of 1, no overlap was found. Our within-sample spillovers were small or non-existent owing to the large physical distance between pairs of sample villages. The minimum straight-line distance between project treatment and control villages was 8.5 miles (13.7 km), which would take at least 2–3 h to traverse by foot. Any spillover benefits largely accrued to others who were not part of the experimental pool.

For a subsample, the research team conducted a follow-up survey to capture knowledge of COVID-19 and COVID-19 vaccines as well as trust in various sources of information. We used data from 878 respondents in 45 villages for which we observed triplets (that is, where we had information on all treatment groups and a 1:1:1 ratio). We collected data from a total of 105 villages (50 control, 30 door-to-door and 25 small-group treatment group villages); however, only for 45 villages did we observe all three treatment groups and therefore provide a clean comparison. Respondents in this subsample of villages were highly similar to those in the overall sample. An overall $F$-test did not reject the equality of means: $P = 0.668$ (Extended Data Table 10).

In treatment villages, these questions were part of the exit survey and implemented 1 day after intervention activities were completed. In control villages, households were visited only once. From a design perspective, we would have ideally captured outcomes at both baseline and endline in each village. It was, however, highly unlikely that these remote places would have been visited by other health personnel from the MoHS or NGOs in the 5-day period between baseline and endline, or that a large number of people would have incurred the cost of visiting the community health clinic for receiving a COVID-19 vaccine. In addition, the costs of revisiting communities in these remote locations are high (the largest line item on the budget relates to transportation costs;

Extended Data Table 9). We verified that there was no vaccination drive conducted during this period. Furthermore, we use the fact that our baseline survey was conducted over a few weeks across communities to inspect the temporal trends in the data. A simple regression of baseline vaccination rates on the date of the baseline survey did not reveal any trend. This reduces the concern that our choice to not revisit control villages affects the conclusions we draw.

Research assistants were blinded with respect to treatment groups and study hypothesis.

### Intervention details
**Timeline of activities.** The research team collaborated closely with the Ministry of Health vaccination team. Both the team of vaccinators and social mobilizers from the MoHS and enumerators in charge of the survey received extensive training on implementation protocols. Only those individuals who were considered proficient after examination were retained for implementation or data collection. Within each village the teams followed several steps outlined below (see Supplementary Fig. 1 for further details). On day 1–2, the research team implemented census listing and baseline surveys described above. On days 3–5, the social mobilizer team engaged in small-group and door-to-door mobilization, the MoHS performed a vaccination drive and the research team conducted exit surveys in treatment villages. On day 6, the research team implemented endline surveys.

**Social mobilization.** The MoHS trained community mobilizers on COVID-19 vaccine safety and efficacy, vaccine types and availability. All mobilizers were trained on how to respond to questions and to counter any misinformation about COVID-19. They were also trained on WHO-recommended safe practices relating to COVID-19 and were instructed to maintain social-distancing protocols and to wear masks when social distancing could not be guaranteed. Additional masks were made available for free for community members.

Community social mobilizers arrived at the village before the mobile vaccination teams. The community mobilizer engaged with local community leaders, including the town chief, section chief, paramount chief, mammy queen, town elders, youth leaders, community health officers, imams, and any other relevant authorities, to seek permission to organize a village information session. The information session took place at a central location, often the community centre or any other convenient location amenable to safe COVID-19 practices.

At the information session, the mobilizer informed community members about COVID-19, available vaccines and evidence about the safety and efficacy of vaccines in preventing transmission and severe illness. People were also informed about the mobile vaccination team and operating procedures during the vaccination drive. They encouraged participants to spread this message to other members of the community not present during the meeting.

In two treatment villages, the MoHS vaccination team did not receive permission from village authorities to conduct the vaccination drive.

**Door-to-door campaign.** In 50 of the 100 villages randomly selected for treatment, community mobilizers approached up to 20 structures randomly selected from the census list, after the group information session was completed. The proportion of each community assigned to treatment therefore varied with the population of the community. In four small communities, all structures were assigned. Owing to logistical complexities and costs, in some communities, mobilizers did not include highly remote village structures (more than 15 min walk from the village centre). This excluded a total of 10 structures (including 40 people aged ≥12 years). Social mobilizers met in private with residents and delivered the same information as was presented at the community meeting. In addition, they addressed people's concerns in private. If the individuals were immediately convinced to get vaccinated, the social mobilizer would guide them to the vaccination site before moving on

to the next household. Neighbours not assigned to receive a home visit were present during the information session in a few cases. In 75% of the communities, these 'compliance issues' were limited to representatives of three or fewer control households, and the majority of communities had no non-compliance of this kind.

**Small-group mobilization.** In the other 50 treatment villages, after the group information session, social mobilizers searched for small groups of people around the village to converse with. Such groups included women washing clothes around the river, individuals gathered at the ataya (tea) shops, residents playing a game of draughts, groups of people around the mosque or church or farm, or residents gathered near the town chief's house. Social mobilizers repeated the same information presented during the community information session. If people inside the small group had already taken the vaccine before this second session, they were invited to talk about their experience. After the session, if residents wanted to take the vaccine, the social mobilizer would guide them to the vaccination site before moving on.

**Mobile vaccination drive.** Vaccines were transported in approved cool boxes or vaccine carriers appropriate for transportation to remote locations. In each treatment village, the MoHS mobile vaccination teams worked with community leaders to select a suitable venue for the vaccination drive. The venue was chosen with the following requirements in mind: it needed to accommodate a waiting area (with some shelter); an arrival and check-in area where patient information can be gathered, maintaining confidentiality; a space for clinical assessment and vaccine administration, including vaccine preparation, maintaining patient confidentiality, privacy and social distancing; an area and system for post-administration observation of patients.

Individuals below 12 years of age were excluded from vaccinations. MoHS teams determined on-site whether a person deemed 'at risk' (for example, pregnant or suffering from severe disease) would also be excluded. After the vaccine was administered, recipients were asked to remain in close proximity to the vaccination team for a minimum of a 15 min in the event that they experienced any unexpected side effect.

Vaccine teams were compliant with MoHS requirements for the storage, preparation, administration and disposal of the vaccine and associated materials. They followed infection prevention and controls and checked the eligibility of people to be vaccinated using the patient checklist.

Mobile teams adhered to MoHS guidelines on informed consent to receive COVID-19 vaccination, ensuring it was taken only by people with the mental capacity to consent to the administration of the vaccines, and taken freely, voluntarily and without coercion. Participants were allowed to withdraw consent at any time.

All vaccine teams received training on vaccinations, including the management of adverse events following immunization. All such events had to be reported using national reporting systems to the MoHS.

### Statistical analysis
To estimate the impact of the intervention on the adult vaccination rate (Extended Data Table 3), we estimated intent-to-treat effects using OLS on individual-level data as follows:

$$Y_{i,j} = \alpha_k + \beta_{1,j} T_{\text{pooled}} + \epsilon_{i,j} \qquad (1)$$

where $Y_{i,j}$ is the vaccination status of individual $i$, in village $j$, $T_{\text{pooled}}$ is the village assignment to either door-to-door or small-group treatment groups, $\alpha_k$ is a vector of randomization block fixed effects (that is, triplet) and $\epsilon_{i,j}$ are heteroscedasticity-robust standard errors clustered at the village level. We estimated effects using a linear estimator (OLS) that accounts for high dimensional fixed effects[49]. In additional analyses, we added to the right-hand side of this equation $Y_{i,j,\text{bl}}$, the baseline vaccination status, and $X_j$, the vector of covariates that

were unbalanced at baseline. We also estimated equation (1) at the village level and for each group, by estimating both $\beta_{1,j}T_{\text{door to door}}$ and $\beta_{2,j}T_{\text{small group}}$ for the door-to-door and small-group treatment groups, respectively (Extended Data Table 1).

To estimate the vaccination count (Extended Data Table 4), we estimated a village-level intent-to-treat effect using OLS on village-level data as follows:

$$Y_j = \alpha_k + \beta_{1,j}T_{\text{pooled}} + \epsilon_j \qquad (2)$$

where $Y_j$ is the number of people vaccinated in village $j$, $T_{\text{pooled}}$ is the village assignment to either door-to-door and small-group treatment groups, $\alpha_k$ is a vector of randomization block fixed effects (that is, triplets) and $\epsilon_j$ is the heteroscedasticity-robust standard error. We estimated equation (2) for several types of respondents. That is, those who were part of the village census, migrants, returnees and those not present during census, and those from other villages, and added $X_j$, a vector of covariates that were unbalanced at baseline.

To assess the individual-level effect of the door-to-door campaign, we restricted our sample to the 50 villages assigned to the door-to-door campaign (that is, $T_{\text{door to door}} = 1$), and estimated intent-to-treat effects using OLS as follows:

$$Y_{i,s} = \alpha_j + \delta_i T_{\text{door to door}} + \mu_{i,s} \qquad (3)$$

where $Y_{i,s}$ is the vaccination status of individual $i$ in structure $s$ (hut or house), $T_{\text{door to door}}$ is the individual-level assignment to receive a visit by the social mobilization team to a structure, $\alpha_j$ is a vector of randomization block fixed effect (that is, the village) and $\mu_{i,s}$ is the heteroscedasticity-robust standard error clustered at the structure level.

For the survey-based outcomes on COVID-19 vaccine knowledge and trust, we estimated equation (1), replacing the dependent variable with the survey responses described above, using the subsample of 45 villages where this dataset was collected and we had data on the full randomization blocks.

For our analysis of the treatment effects by subgroup, we estimated equation (1) separately for men, women, various age groups (18–24 years, 25–54 years and >55 years), and sample splits based on whether the household head had any schooling, owns any land or reduced portions of food. To test for differences across subgroups, we estimate equation (1) and interact the subgroup variable with treatment.

In the results presented in Extended Data Tables 6 and 7, we also adjusted for the fact that we conducted multiple tests on the same dataset by implementing FDR corrections and report the FDR $q$ values[50]. We also report the bootstrapped $P$ value[51] to account for regressions with a small number of clusters.

## Literature review of vaccination uptake RCT studies

We conducted a literature review of articles in PubMed published between 1 January 2000 and 7 January 2023 using the search terms '(vaccin*[Title/Abstract] OR immun*[Title/Abstract]) AND *additional search term*[Title/Abstract]) AND (Randomized Controlled Trial[Publication Type])', with the following additional search terms: 'access'; 'community-based'; 'cost effect*'; 'demand'; 'hesitant'; 'incentive*'; 'intervention*'; 'mobile'; 'nudge*'; 'rural'; and 'supply'. These searches returned 3,615 unique articles. We screened out articles that were not related to vaccine uptake or that did not use a RCT, which reduced the sample to 141 articles. We appended a further 20 relevant studies that were identified by snowballing and rejected 17 papers that did not have a control group, did not report the percentage-point change in vaccine uptake or did not include a test statistic. The final list of 144 articles comprises 234 distinct interventions for which we can report a percentage-point change relative to a control group

(Supplementary Table 2). Of these, 33 interventions (14%) reported information about the cost of the intervention per vaccine administered. This cost specifically refers to the cost of implementing the intervention and does not include the cost of the vaccine itself. Studies that did not unequivocally state the cost of the intervention per vaccinated person were not included in our cost-effectiveness comparisons. Two studies reported the cost in currencies other than US dollars[52,53], and these costs were converted to the US dollar equivalent for the year the study was published, using exchange rate data from the respective countries' national statistics agencies. We did not analyse publication bias.

## Deviations from pre-registered hypotheses

We pre-registered our research protocol and hypotheses at the ISRCTN registry (study ISRCTN17878735).

We report on our main hypothesis in Fig. 2 and Extended Data Table 3. In addition to reporting on our main pre-registered outcome (adult vaccination rate), we also report on the total immunizations given per vaccination site because many more people showed up to our temporary clinics from neighbouring villages or were not present during the pre-intervention census, and we had not anticipated this. Figure 3 and Extended Data Table 4 therefore report on the count of all individuals (aged 12 years and above) who visited our clinics to receive a vaccination. This metric is necessary to correctly compute the cost-effectiveness.

The heterogeneity analysis reported in Fig. 5, in which we analysed whether vaccination rates differ by age, sex, schooling and wealth variables, was not pre-specified and followed heterogeneity tests that are common in the vaccine literature[54].

## Reporting summary

Further information on research design is available in the Nature Portfolio Reporting Summary linked to this article.

## Data availability

The primary survey data forming part of this study were collected using SurveyCTO software (v.2.81). These de-identified datasets are available in the Harvard Dataverse at https://doi.org/10.7910/DVN/PRXF5Z.

## Code availability

All analysis for this paper was conducted using Stata SE 17. Replication files and de-identified data are available in the Harvard Dataverse at https://doi.org/10.7910/DVN/PRXF5Z.

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

**Acknowledgements** Funding support for this research was provided by Weiss Asset Management, the Dutch Research Council (NWO) (VI.Vidi.191.154), UKRI and the International Growth Centre (SLE - 21148). We are indebted and thankful to study participants for giving their time; O. Sawaneh, A. Turray and our team of enumerators for excellent research assistance; J. W. Ansumana, A. K. Kemoh, N. S. Kamara, T. Boima and the wider team the MoHS of the Government of Sierra Leone; the district organization officers for the districts we went to (A. S. Conteh, A. Sesay, J. Kalokoh, M. Sesay, L. Mansaray, I. Bangura, D. Kanu and A. D. D. Sesay);

and the team at Concern Worldwide, Yale Research Initiative on Innovation and Scale (Y-RISE) and the International Growth Centre.

**Author contributions** N.F.M., A.M.M., M.V., S.C. and D.M.K. conceptualized the project. V.R. and M.L. curated the data. V.R., M.L., E.T., M.R. and M.V. undertook formal analyses. N.F.M., A.M.M. and M.V. acquired funding. N.F.M., M.L., M.R., S.C., S.K. and J.N. performed the investigations. N.F.M., A.M.M., M.V., V.R and M.L. designed the methodology. N.F.M. and M.V. administrated the project. M.L., V.R., S.K., J.N. and M.R. supervised the work. All authors validated the findings. V.R., E.T., M.V. and M.R. visualized the data. A.M.M. and M.V. wrote the first draft, with all authors contributing to writing, and all authors contributed to the review and editing of the paper.

**Competing interests** The authors declare no competing interests.

## Additional information

**Correspondence and requests for materials** should be addressed to Niccolò F. Meriggi or Ahmed Mushfiq Mobarak.

**Extended Data Table 1 | Intent-to-Treat Effect of Door-to-Door and Small-Group Treatments**

|  | (1) | (2) | (3) |
|---|---|---|---|
| Door-to-Door | 0.293*** | 0.279*** | -0.011 |
|  | (0.023) | (0.023) | (0.019) |
| Small-Group | 0.231*** | 0.235*** |  |
|  | (0.021) | (0.020) |  |
| Proportion Vaccinated at Baseline |  | 0.628*** |  |
|  |  | (0.076) |  |
| Additional Covariates | No | Yes | No |
| Observations | 12096 | 12096 | 3760 |
| Mean in Control | 0.062 | 0.062 | 0.282 |
| No. of Villages | 150 | 150 | 50 |
| No. of Structures | 3479 | 3479 | 1120 |
| $P(\_Door\text{-}to\text{-}Door = \_Small\text{-}Group)$ | 0.014 | 0.042 |  |
| $R^2$ | 0.137 | 0.157 | 0.140 |

The table presents Intent-To-Treat estimates of the adult vaccination rate. Column 1 presents the adult vaccination rate for the sub-treatment arms. Treatment effects are estimated using OLS and include randomization block fixed effects (triplets) and heteroscedasticity-robust standard errors clustered at the village level. Column 2 adds covariates that were imbalanced at baseline. Column 3 presents the results of the individual Door-to-Door campaign where up to 20 structures were randomly assigned to be visited by the social mobilization team. The sample is restricted to the 50 villages assigned to the Door-to-Door treatment arm, and non-peripheral structures within these villages. Treatment effects are estimated using OLS including randomization block fixed effects (villages), with heteroscedasticity-robust standard errors clustered at the structure (hut, house) level. The p-value in the bottom panel is from a two-sided t-test on the quality of means of both treatment arms. ***, **, and * indicate significance at the 1, 5, and 10 percent critical level and refer to two-sided tests without multiple comparison adjustments.

## Extended Data Table 2 | Baseline Descriptive Statistics and Statistical Balance

| variable | Control Mean (SD) | Control-Door-to-Door Diff(SE) | Control-Small-Group Diff (SE) | Door-to-Door-Small-Group Diff (SE) | N |
|---|---|---|---|---|---|
| **Community Characteristics from 2015 Census** | | | | | |
| **Percent of infants in locality fully immunized** | 0.502 | 0.015 | -0.016 | 0.031 | 150 |
| | (0.300) | (0.034) | (0.034) | (0.035) | |
| **Percent of locality that is literate** | 0.284 | -0.028 | -0.016 | -0.013 | 150 |
| | (0.206) | (0.023) | (0.028) | (0.026) | |
| **Average age** | 21.483 | -0.118 | 0.053 | -0.171 | 150 |
| | (2.366) | (0.321) | (0.308) | (0.341) | |
| **Lives five miles or more from health facility** | 0.960 | 0.022 | -0.000 | 0.022 | 150 |
| | (0.196) | (0.018) | (0.011) | (0.018) | |
| **Percent of locality that is Christian** | 0.119 | -0.005 | 0.027 | -0.032 | 150 |
| | (0.249) | (0.038) | (0.044) | (0.044) | |
| **Percent of locality that is Muslim** | 0.867 | 0.010 | -0.014 | 0.024 | 150 |
| | (0.261) | (0.040) | (0.045) | (0.044) | |
| **Percent of locality born in the same Chiefdom** | 0.931 | -0.020 | 0.008 | -0.028 | 150 |
| | (0.115) | (0.022) | (0.020) | (0.023) | |
| **Percent of locality that is employed in agriculture.** | 0.937 | -0.076 | -0.052 | -0.024 | 150 |
| | (0.078) | (0.032) | (0.029) | (0.032) | |
| **Percent of locality with Access to Internet** | 0.027 | -0.010 | -0.016 | 0.007 | 150 |
| | (0.073) | (0.011) | (0.010) | (0.008) | |
| **Living in formal structures** | 0.795 | 0.046 | 0.003 | 0.043 | 150 |
| | (0.354) | (0.056) | (0.059) | (0.056) | |
| **Own land** | 0.989 | -0.022 | -0.023 | 0.001 | 150 |
| | (0.023) | (0.017) | (0.016) | (0.019) | |
| **Lives within five miles of primary school** | 0.616 | -0.117 | -0.019 | -0.098 | 150 |
| | (0.478) | (0.081) | (0.078) | (0.080) | |
| **Lives within five miles of water source** | 0.942 | -0.041 | -0.047 | 0.006 | 150 |
| | (0.211) | (0.051) | (0.049) | (0.054) | |
| **Owns a radio** | 0.508 | -0.105 | -0.080 | -0.024 | 150 |
| | (0.293) | (0.054) | (0.055) | (0.051) | |
| **Owns a cell phone** | 0.315 | -0.026 | -0.020 | -0.005 | 150 |
| | (0.249) | (0.048) | (0.047) | (0.046) | |
| **With formal roofs** | 0.546 | 0.055 | 0.060 | -0.005 | 150 |
| | (0.365) | (0.051) | (0.059) | (0.058) | |
| **Average number of assets** | 1.704 | -0.112 | -0.063 | -0.049 | 150 |
| | (0.699) | (0.159) | (0.150) | (0.146) | |
| **Characteristics from Village Census** | | | | | |
| **Village population** | 187.080 | -5.400 | 35.920 | -41.320 | 150 |

The table presents baseline data for the 150 study communities using 2015 census and the village census. Column (1) shows the mean and standard deviation for the control group. Columns (2) and (3) display regression coefficients and standard errors of the Door-to-Door and Small-Group treatment arms compared to the control group. Column (4) indicates differences between the two treatment arms. Column (5) shows the number of communities included in the regression. Regressions include randomization block fixed effects and heteroscedasticity-robust standard errors. All the measures are constructed from household level data aggregated to the community level. The last row reports p-values from two-sided Joint Orthogonality tests, from a multinomial logit regression with the treatment indicator as the dependent variable, regressed on all the variables in the table.

**Extended Data Table 3 | Intent-to-treat Estimates of Vaccination Rate of People Enumerated During Census**

|  | (1) | (2) | (3) |
|---|---|---|---|
| Pooled Treatment | 0.261*** | 0.254*** | 0.283*** |
|  | (0.018) | (0.019) | (0.025) |
| Proportion Vaccinated at Baseline |  | 0.659*** |  |
|  |  | (0.076) |  |
| Additional Covariates | No | Yes | No |
| Bootstrapped P-Value | 0.000 | 0.000 | 0.000 |
| Mean in Control | 0.062 | 0.062 | 0.061 |
| No. of Observations | 12096 | 12096 | 150 |
| No. of Villages | 150 | 150 | 150 |
| $R^2$ | 0.13 | 0.16 | 0.66 |

This table presents Intent-To-Treat estimates corresponding to Fig. 2. The dependent variable in Columns (1) and (2) is the individual level vaccination status at endline. Treatment effects are estimated using OLS including randomization fixed effects (ie for each triplet) and heteroscedasticity-robust standard errors clustered at the community level. Included covariates in Column (2) are: the baseline adult vaccination rate; proportion of households employed in agriculture; proportion of households that own a radio; the proportion of women breastfeeding and proportion of households that own land. In Column (3), the dependent variable is the proportion of adults vaccinated in each community. ***, **, and * indicate significance at the 1, 5, and 10 percent critical level and refer to two-sided tests without multiple comparison adjustments. Bootstrapped p-value is the p-value resulting from a wild bootstrap test of Pooled Treatment = 0, with 999 repetitions.

**Extended Data Table 4 | Intent-To-Treat estimates of the Count of People Vaccinated per Site After Mobile Vaccination Program**

|  | (1) | (2) | (3) | (4) | (5) | (6) |
|---|---|---|---|---|---|---|
| Pooled treatment | 26.950[***] | 39.250[***] | 50.200[***] | 26.138[***] | 39.859[***] | 50.532[***] |
|  | (2.532) | (3.622) | (4.018) | (2.929) | (4.414) | (4.708) |
|  |  |  |  |  |  |  |
| Proportion Vaccinated at Baseline |  |  |  | 36.897[***] | 43.539[***] | 42.348[**] |
|  |  |  |  | (13.418) | (16.511) | (17.867) |
| Additional Covariates | No | No | No | Yes | Yes | Yes |
| Bootstrapped P-Value | 0.000 | 0.000 | 0.000 | 0.000 | 0.000 | 0.000 |
| Mean in Control | 5.060 | 5.060 | 5.060 | 5.060 | 5.060 | 5.060 |
| No. of Observations | 150 | 150 | 150 | 150 | 150 | 150 |
| R squared | 0.59 | 0.58 | 0.65 | 0.64 | 0.62 | 0.68 |

This table presents Intent-To-Treat estimates corresponding to Fig. 3. Treatment effects are estimated using OLS including randomization fixed effects (ie for each triplet) and heteroscedasticity-robust standard errors clustered at the community level. The dependent variable is the count of people vaccinated by the end of the study. In Column (1) we restrict our sample to those enumerated during the census. In Column (2) we add people who travelled from other communities for vaccination. In Column (3) we do not restrict our sample. Columns (4)-(6) add covariates imbalanced at baseline. Included covariates are: the baseline adult vaccination rate; proportion of households employed in agriculture; proportion of households that own a radio; the proportion of women breastfeeding and proportion of households that own land. ***, **, and * indicate significance at the 1, 5, and 10 percent critical level and refer to two-sided tests without multiple comparison adjustments. Bootstrapped p-value is the p-value resulting from a wild bootstrap test of Pooled Treatment = 0, with 999 repetitions.

**Extended Data Table 5 | Proportion Vaccinated by Baseline Willingness to Take Vaccines and Meeting Attendance**

| | HH member attended meeting | | |
| --- | --- | --- | --- |
| | No (1,066) | Yes (636) | Total |
| \textbf{Would take COVID-19 vaccine if offered} | | | |
| No (279) | 0.144 | 0.538 | 0.272 |
| Yes (1,423) | 0.394 | 0.646 | 0.491 |
| Total | 0.350 | 0.631 | 0.455 |

Each cell indicates the vaccination rate for adults surveyed in the baseline by whether they attended the village meeting crossed by whether they indicated if they were willing to take the COVID-19 vaccine during the course of the pre-meeting baseline survey.

**Extended Data Table 6 | Intent-To-Treat estimates for Knowledge and Attitudes Towards Vaccines in Sub-sample**

| | (1) Believes COVID-19 is real | (2) Knows about the COVID-19 vaccine | (3) Believes vaccines are effective | (4) Believes vaccines are safe |
|---|---|---|---|---|
| Pooled Treatment | 0.051 (0.035) | 0.108** (0.044) | 0.097 (0.074) | 0.131* (0.070) |
| Bootstrapped P-Value | 0.274 | 0.070 | 0.311 | 0.153 |
| FDR Q-value | 0.162 | 0.084 | 0.162 | 0.117 |
| Mean in Control | 0.876 | 0.776 | 0.267 | 0.244 |
| No. of Observations | 817 | 817 | 686 | 686 |
| No. of Villages | 45 | 45 | 45 | 45 |
| $R^2$ | 0.09 | 0.12 | 0.14 | 0.14 |

This table presents Intent-To-Treat estimates of the corresponding to Fig. 4. The dependent variables are indicators of knowledge and attitudes of COVID and COVID vaccines, included in Fig. 4. Treatment effects are estimated using OLS including randomization fixed effects (ie for each triplet) and heteroscedasticity-robust standard errors clustered at the community level. Sub-sample comprises 45 villages and 817 households surveyed at endline. ***, **, and * indicate significance at the 1, 5, and 10 percent critical level and refer to two-sided tests. Bootstrapped p-value is the p-value resulting from a wild bootstrap test of Pooled Treatment = 0, with 999 repetitions. FDR q-values are included to account for testing across several outcomes.

**Extended Data Table 7 | Intent-To-Treat estimates for Which Source People Trust Most for Information on COVID-19 in Sub-sample**

| | (1) Community Health Clinic | (2) Ministry of Health and Sanitation | (3) Media | (4) Social media | (5) Family and friends |
|---|---|---|---|---|---|
| Pooled Treatment | 0.013 (0.059) | 0.011 (0.025) | -0.028 (0.047) | -0.002 (0.004) | -0.026 (0.022) |
| Bootstrapped P-Value | 0.855 | 0.707 | 0.604 | 0.626 | 0.375 |
| FDR Q-value | . | . | . | . | . |
| Mean in Control | 0.213 | 0.066 | 0.290 | 0.011 | 0.066 |
| No. of Observations | 817 | 817 | 817 | 817 | 817 |
| No. of Villages | 45 | 45 | 45 | 45 | 45 |
| $R^2$ | 0.07 | 0.11 | 0.25 | 0.04 | 0.08 |

This table presents Intent-To-Treat estimates of the corresponding to Fig. 4. The dependent variables are indicators for which source respondents trust for information relating to COVID-19, included in Fig. 4. Treatment effects are estimated using OLS including randomization fixed effects (ie for each triplet) and heteroscedasticity-robust standard errors clustered at the community level. Trust indicators are constructed from the household level endline. Sub-sample comprises 45 villages and 817 households surveyed at endline. ***, **, and * indicate significance at the 1, 5, and 10 percent critical level and refer to two-sided tests. Bootstrapped p-value is the p-value resulting from a wild bootstrap test of Pooled Treatment = 0, with 999 repetitions. FDR q-values are included to account for testing across several outcomes.

**Extended Data Table 8 | Intent-To-Treat Estimates for Demographic Sub-groups**

| | (1) Full sample | (2) Female | (3) Male | (4) Aged 18-24 | (5) Aged 25-54 | (6) Aged 55+ | (7) HH head any schooling | (8) HH head no schooling | (9) HH owns any land | (10) HH owns no land | (11) HH reduced food portions | (12) HH did not reduce food portions |
|---|---|---|---|---|---|---|---|---|---|---|---|---|
| Pooled Treatment | 0.261*** | 0.231*** | 0.301*** | 0.181*** | 0.274*** | 0.316*** | 0.234*** | 0.262*** | 0.402*** | 0.372*** | 0.380*** | 0.391*** |
| | (0.018) | (0.020) | (0.020) | (0.018) | (0.019) | (0.028) | (0.024) | (0.019) | (0.028) | (0.028) | (0.032) | (0.028) |
| Mean in Control | 0.062 | 0.056 | 0.070 | 0.036 | 0.064 | 0.094 | 0.059 | 0.063 | 0.106 | 0.073 | 0.115 | 0.084 |
| No. of Observations | 12096 | 6797 | 5299 | 2662 | 7512 | 1922 | 2582 | 9514 | 1761 | 913 | 1072 | 1602 |
| No. of Villages | 150 | 150 | 150 | 150 | 150 | 145 | 139 | 150 | 147 | 144 | 139 | 149 |
| R squared | 0.13 | 0.13 | 0.16 | 0.11 | 0.14 | 0.20 | 0.15 | 0.14 | 0.23 | 0.24 | 0.22 | 0.24 |

This table presents Intent-To-Treat estimates of the pooled treatment for demographic sub-groups included in Fig. 5. Treatment effects are estimated using OLS including randomization fixed effects (ie for each triplet) and heteroscedasticity-robust standard errors clustered at the community level. Dependent variable is the vaccination status at the end of the study of adults enumerated during the census. ***, **, and * indicate significance at the 1, 5, and 10 percent critical level and refer to two-sided tests without multiple comparison adjustments.

**Extended Data Table 9 | Cost-Effectiveness Analysis**

|  | Cost Type | Total Cost | Cost Per Vaccination | N |
|---|---|---|---|---|
| Training Costs |  |  |  |  |
| Training venue | Fixed | 14610.6 | 3.07 | 4771 |
| DSA for trainees (76 team members + 7 DOOs) | Fixed | 14094.8 | 2.96 | 4771 |
|  |  |  |  |  |
| Debriefing |  |  |  |  |
| Training venue | Fixed | 5844.3 | 1.23 | 4771 |
| DSA for trainees (76 team members + 7 DOOs) | Fixed | 7047.4 | 1.48 | 4771 |
|  |  |  |  |  |
| Materials |  |  |  |  |
| Printing of vaccination cards | Variable | 6809.1 | 1.43 | 4771 |
| Printing of screening forms | Variable | 8511.4 | 1.79 | 4771 |
|  |  |  |  |  |
| Transport/Communication |  |  |  |  |
| Vehicle hire and fuel | Variable | 47401.5 | 9.95 | 4771 |
| Fuel for DOOs | Variable | 7507.0 | 1.58 | 4771 |
| Mobile phone top-up (per team) | Variable | 289.4 | 0.06 | 4771 |
|  |  |  |  |  |
| Salaries Vaccination Teams |  |  |  |  |
| Daily rate for vaccinators | Variable | 13661.8 | 2.87 | 4771 |
| DSA for vaccinators | Variable | 25615.8 | 5.38 | 4771 |
|  |  |  |  |  |
| Total |  | 151393.0 | 31.77 | 4771 |

Author calculations based on implementation budget and financial reports from implementing partners.

## Extended Data Table 10 | Comparison of Full Sample to Sub-sample

| variable | N | Full Sample Mean (SE) | Restricted Sample N | Restricted Sample Mean (SE) | Difference (p-value) |
|---|---|---|---|---|---|
| **Community Characteristics from 2015 Census** | | | | | |
| Proportion of infants in locality fully immunized | 150 | 0.502 | 45 | 0.566 | -0.064 |
| | | (0.293) | | (0.283) | (0.078) |
| Proportion of locality that is literate | 150 | 0.269 | 45 | 0.265 | 0.004 |
| | | (0.192) | | (0.191) | (0.877) |
| Average age | 150 | 21.462 | 45 | 21.009 | 0.453 |
| | | (2.648) | | (3.135) | (0.171) |
| Lives five miles or more from health facility | 150 | 0.968 | 45 | 0.955 | 0.013 |
| | | (0.174) | | (0.208) | (0.560) |
| Proportion of locality that is Christian | 150 | 0.126 | 45 | 0.099 | 0.028 |
| | | (0.253) | | (0.209) | (0.384) |
| Proportion of locality that is Muslim | 150 | 0.866 | 45 | 0.886 | -0.021 |
| | | (0.257) | | (0.226) | (0.523) |
| Proportion of locality born in the same Chiefdom | 150 | 0.927 | 45 | 0.948 | -0.021 |
| | | (0.125) | | (0.105) | (0.177) |
| Proportion of locality that is employed in agriculture. | 150 | 0.894 | 45 | 0.894 | 0.000 |
| | | (0.199) | | (0.210) | (0.987) |
| Proportion of locality with Access to Internet | 150 | 0.018 | 45 | 0.010 | 0.008 |
| | | (0.048) | | (0.017) | (0.177) |
| Living in formal structures | 150 | 0.812 | 45 | 0.756 | 0.055 |
| | | (0.330) | | (0.352) | (0.181) |
| Own land | 150 | 0.973 | 45 | 0.972 | 0.001 |
| | | (0.087) | | (0.104) | (0.913) |
| Lives within five miles of primary school | 150 | 0.570 | 45 | 0.561 | 0.009 |
| | | (0.477) | | (0.479) | (0.878) |
| Lives within five miles of water source | 150 | 0.913 | 45 | 0.909 | 0.005 |
| | | (0.258) | | (0.265) | (0.886) |
| Owns a radio | 150 | 0.446 | 45 | 0.421 | 0.026 |
| | | (0.284) | | (0.288) | (0.473) |
| Owns a cell phone | 150 | 0.300 | 45 | 0.289 | 0.011 |
| | | (0.253) | | (0.255) | (0.722) |
| With formal roofs | 150 | 0.584 | 45 | 0.511 | 0.074 |
| | | (0.350) | | (0.341) | (0.092) |
| Average number of assets | 150 | 1.646 | 45 | 1.650 | -0.005 |
| | | (0.766) | | (0.714) | (0.962) |
| **Characteristics from Village Census** | | | | | |
| Village population | 150 | 197.253 | 45 | 203.222 | -5.969 |
| | | (120.936) | | (126.285) | (0.694) |
| Proportion of adults vaccinated at baseline | 150 | 0.092 | 45 | 0.091 | 0.001 |
| | | (0.131) | | (0.143) | (0.955) |
| HH head has had any schooling | 150 | 0.184 | 45 | 0.165 | 0.018 |
| | | (0.117) | | (0.109) | (0.208) |
| Owns land | 149 | 0.647 | 45 | 0.623 | 0.025 |
| | | (0.214) | | (0.228) | (0.354) |
| Reduced portion sizes in last week | 149 | 0.391 | 45 | 0.364 | 0.027 |
| | | (0.221) | | (0.219) | (0.329) |
| Age | 150 | 22.458 | 45 | 22.210 | 0.248 |
| | | (2.639) | | (2.481) | (0.453) |
| HH head is female | 150 | 0.259 | 45 | 0.298 | -0.039 |
| | | (0.119) | | (0.110) | (0.009) |
| Is breastfeeding | 150 | 0.133 | 45 | 0.134 | -0.001 |
| | | (0.060) | | (0.067) | (0.902) |
| Is pregnant | 150 | 0.041 | 45 | 0.033 | 0.008 |
| | | (0.055) | | (0.027) | (0.225) |
| Joint F-test p-value | | 0.668 | | | |

This table presents baseline balance data for the full sample of 150 villages and the 45 villages in the restricted subsample used in Extended Data Tables 6 and 7. Column (1) and (3) report the complete and restricted sample. Column (2) and (4) report the variable mean and standard error for each sample. Column (5) reports the mean difference. The last row reports the p-value from an F-test, from a regression with the subsample indicator as the dependent variable, regressed on all the variables in the table.

# Reporting Summary

## Statistics

For all statistical analyses, confirm that the following items are present in the figure legend, table legend, main text, or Methods section.

| n/a | Confirmed | |
|---|---|---|
| ☐ | ☒ | The exact sample size (*n*) for each experimental group/condition, given as a discrete number and unit of measurement |
| ☐ | ☒ | A statement on whether measurements were taken from distinct samples or whether the same sample was measured repeatedly |
| ☐ | ☒ | The statistical test(s) used AND whether they are one- or two-sided *Only common tests should be described solely by name; describe more complex techniques in the Methods section.* |
| ☐ | ☒ | A description of all covariates tested |
| ☐ | ☒ | A description of any assumptions or corrections, such as tests of normality and adjustment for multiple comparisons |
| ☐ | ☒ | A full description of the statistical parameters including central tendency (e.g. means) or other basic estimates (e.g. regression coefficient) AND variation (e.g. standard deviation) or associated estimates of uncertainty (e.g. confidence intervals) |
| ☐ | ☒ | For null hypothesis testing, the test statistic (e.g. *F*, *t*, *r*) with confidence intervals, effect sizes, degrees of freedom and *P* value noted *Give P values as exact values whenever suitable.* |
| ☒ | ☐ | For Bayesian analysis, information on the choice of priors and Markov chain Monte Carlo settings |
| ☒ | ☐ | For hierarchical and complex designs, identification of the appropriate level for tests and full reporting of outcomes |
| ☐ | ☒ | Estimates of effect sizes (e.g. Cohen's *d*, Pearson's *r*), indicating how they were calculated |

*Our web collection on statistics for biologists contains articles on many of the points above.*

## Software and code

Policy information about availability of computer code

| Data collection | For primary data collection we used ODK survey collection software, version 2.81 |
|---|---|
| Data analysis | For our data analysis we STATA SE 17 |

For manuscripts utilizing custom algorithms or software that are central to the research but not yet described in published literature, software must be made available to editors and reviewers. We strongly encourage code deposition in a community repository (e.g. GitHub). See the Nature Portfolio guidelines for submitting code & software for further information.

## Data

Policy information about availability of data

All manuscripts must include a data availability statement. This statement should provide the following information, where applicable:

- Accession codes, unique identifiers, or web links for publicly available datasets
- A description of any restrictions on data availability
- For clinical datasets or third party data, please ensure that the statement adheres to our policy

Primary survey data was collected for the research and can be made available to reviewers.
All survey will be de-identified and deposited in Harvard Dataverse upon publication.

# Research involving human participants, their data, or biological material

Policy information about studies with human participants or human data. See also policy information about sex, gender (identity/presentation), and sexual orientation and race, ethnicity and racism.

| | |
|---|---|
| Reporting on sex and gender | We include subgroup analysis by gender (as self reported by respondents) in Figure 5 and Extended Data Table 8 |
| Reporting on race, ethnicity, or other socially relevant groupings | We also provide subgroup analysis by education (any education, or none), age group (18-25, 25-54, and older) and land ownership (whether the household owns any land), food security (whether households reduced portions of food) in in Figure 5 and Extended Data Table 8 |
| Population characteristics | The population of these villages was on average 22.3 years old, 26.5% of households were female-headed, 64.7% of people lived in a household of 6 or fewer people. Just 20.2% lived in a household where the household head had any form of formal schooling, and about 86.2% lived in a household where the head was primarily engaged in farming. |
| Recruitment | Participant inclusion criteria for the intervention: All residents of selected rural communities eligible for the COVID-19 vaccine in 150 communities selected from seven rural districts in Sierra Leone (Koinadugu, Falaba, Karene, Kambia, Tonkolili, Bombali and Port Loko). For recruitment into the survey, see Sampling Strategy below. |
| Ethics oversight | We received Institutional Review Board (IRB) approval from the Sierra Leone Ethics and Scientific Review Committee (SLERC 20220210), Yale University (2000031541) and Wageningen University (WUR 20220222). The research protocol was pre-registered at ISRCTN (study ISRCTN 17878735, see https://doi.org/10.1186/ISRCTN17878735). All study participants completed informed consent. |

Note that full information on the approval of the study protocol must also be provided in the manuscript.

# Field-specific reporting

Please select the one below that is the best fit for your research. If you are not sure, read the appropriate sections before making your selection.

☐ Life sciences   ☒ Behavioural & social sciences   ☐ Ecological, evolutionary & environmental sciences

For a reference copy of the document with all sections, see nature.com/documents/nr-reporting-summary-flat.pdf

# Behavioural & social sciences study design

All studies must disclose on these points even when the disclosure is negative.

| | |
|---|---|
| Study description | We conduct a randomized control trial where we study the impact of an intervention increasing access to vaccines (specifically COVID-19 vaccines) and two different mobalisation strategies on vaccine uptake in rural communities in Sierra Leone, a lower income country characterized by a poor health infrastructure. We find that our intervention increased vaccine uptake by over 25 pp and can be implemented quickly and cost effectively. These features make our intervention particularly suited for situations where a quick vaccination roll out is needed in places that are severely resource constrained. |
| Research sample | The objective of the study was to learn about last mile challenges in vaccine uptake in Low Income Countries. The sample consists of 150 rural and remote villages in 7 districts (Koinadugu, Falaba, Karene, Kambia, Tonkolili, Bombali, Port Loko) in Sierra Leone. Villages lie outside the regular catchment areas of health clinics, where access to vaccines are is low, -- ie "hard cases". The majority of sub-Saharan Africans reside in rural areas, and face similar challenges as our study sample. |
| Sampling strategy | We used a power calculation to determine the number of villages needed for a range of minimum detectable effect sizes assuming a 5% significance level with 80% power. We expected decisions to take a vaccine should be highly correlated within a village, so we assumed an ICC of 0.15. In the 2015 census data, the mean population is 2480 people per village. Assuming that 50% of them were eligible to take the vaccine, we assumed an effective village size of 1200 people per village. Given that vaccine take up in Sierra Leone is low, we assumed a baseline vaccination rate of 2.5%. Based on the treatment effects reported in the literature for similar studies and the large heterogeneity in treatment effects reported, we adopt a conservative approach and set our expected MDE at 0.05. We oversampled slightly and the final design included 150 communities across the three treatment arms, in 1:1:1 ratio.

Village level sample: we used the 2015 Sierra Leone Census and selected 7 largely rural districts (Koinadugu, Falaba, Karene, Kambia, Tonkolili, Bombali, Port Loko) with 8,784 communities in 54 Chiefdoms. We then restricted the sample to communities with no health clinic within five miles of the community center, resulting in 1,849 communities. We further excluded very small communities (below 19 structures) and for which latitude and longitude was missing. The final sampling frame of 420 communities were located in 49 Chiefdoms and 7 Districts. Within each District, we then matched communities on the following strata: i. the share of the population that was immunized, ii. the age of the population, iii. literacy levels, and the iv. distance from the closest clinic. This allowed us to identify communities that had the most similar characteristics within a district and used this to assign the most similar communities to one of the treatment group and establish comparable "triplets". This resulted in 106 triplets in total. We then randomly selected 50 triplets (and 150 villages) using district as a blocking variable. The final list included: 9 triplets each for Koinadugu and Falaba District, 8 triplets for Karene District and 6 triplets each for Port Loko, Tonkolili, Kambia and Bombali District. Within each village we used several instruments collect data:
• Census with all household heads, going door to door, to help make a roster of those living under the same roof and eating from the |

same pot.
• Baseline household survey:  n all communities, we randomly selected a sample of 30 households per village from the census/listing exercise
• Exit survey: of those that took a vaccine taken at each mobile vaccination site
• Endline household survey: with the same households that we interviewed at baseline

| | |
|---|---|
| Data collection | Data was collected using a standardized questionnaire on tablets using  SurveyCTO software. Questionnaires were conducted by local research assistants hired by the research team, using electronic tablets to record survey responses. Besides these research assistants, no one from the research team or from the implementing partner was present during the questionnaires. The research assistants were trained to conduct the questionnaire but were blinded to the research questions of this study. Other aspects of training including following important ethical guidelines, maintaining neutrality, and ensuring the respondents' comfort during the process. High frequency checks were implemented throughout the data collection process. The research assistants' work was routinely audited to ensure consistent adherence to the standardized questionnaire and data collection protocols. Besides quality checks, survey data was stored in a password-protected server, and was only accessible to the authors of this submission. |
| Timing | Our survey data were collected, and the accompanying interventions undertaken, during March and April 2022. |
| Data exclusions | No data were excluded from analysis. |
| Non-participation | All 150 village remained in the study throughout. At the respondent level: for the baseline 4 respondents, endline 0 respondents and for the exit survey 6 respondents declined participation. No reasons were for non consent were provided. |
| Randomization | We used blocked randomisation: within each block of three villages we assigned  villages to either a control group, "door to door" treatment, or "small group" treatment arm. in a 1:1:1 ratio. This resulted in 50 villages assigned to control, 50 to "door to door", and 50 to "small group". Within the "door to door" villages, we randomly assigned randomly selected 20 residential structures from the community census list to receive a visit from the social mobilisation team. |

# Reporting for specific materials, systems and methods

We require information from authors about some types of materials, experimental systems and methods used in many studies. Here, indicate whether each material, system or method listed is relevant to your study. If you are not sure if a list item applies to your research, read the appropriate section before selecting a response.

## Materials & experimental systems

| n/a | Involved in the study |
|---|---|
| ☒ | ☐ Antibodies |
| ☒ | ☐ Eukaryotic cell lines |
| ☒ | ☐ Palaeontology and archaeology |
| ☒ | ☐ Animals and other organisms |
| ☒ | ☐ Clinical data |
| ☒ | ☐ Dual use research of concern |
| ☒ | ☐ Plants |

## Methods

| n/a | Involved in the study |
|---|---|
| ☒ | ☐ ChIP-seq |
| ☒ | ☐ Flow cytometry |
| ☒ | ☐ MRI-based neuroimaging |

