## [Peer Review File · Nature]

Manuscript Title: Last-Mile Delivery Increases Vaccine Uptake in Sierra Leone

Reviewer Comments & Author Rebuttals

Reviewer Reports on the Initial Version:

Referees' comments:

Referee #1 (Remarks to the Author):

Review of "Solving Last-Mile Delivery Challenges is Critical to Increase COVID-19 Vaccine Uptake: A Cluster Randomized Controlled Trial"
Nature 2022-09-14388

A. Summary of the key results

This paper shows how a novel collaboration between the government of Sierra Leone and the NGO Concern Worldwide and social scientists dramatically increased

COVID vaccinations in the most remote and rural areas in Sierra Leone where vaccination rates had been low. The project deployed a set of complementary interventions in each village including: physically bringing vaccines and medical personnel to administer it to villages, establishing relationships with

village authorities before connecting with the broader village, hosting community wide meetings attended by and endorsed by the authorities, and deploying local mobilizers to walk around villages with information and encouragement. This project also incorporated a rigorous evaluation of its impact and an impartial allocation of the project to villages given what I imagine must have been tight budget constraints: since the intervention had to

be costly and the government and NGO could not provide it to all villages, the

set of interventions was randomly assigned to certain villages and not to others. This impact evaluation convinces us that this approach (and not some

other event) caused villagers in the villages assigned to the bundle of services

and outreach to vaccinate themselves at much higher rate than villagers in villages assigned to the status quo condition the within a 2--3 day period.

Although I have seen collaborations between researchers and governments to use

SMS messages and postcards to increase vaccination rates in the USA and to embed

randomized trials into such interventions to assess their impact, I have not

seen an approach that involves the physical transportation of an entire bundle

of interventions to remote populations in the Global South combined with a randomized evaluation, nor a collaboration between a government, an NGO and

a team of social scientists. Compared to light-touch approaches to encouraging

vaccination common outside the Global South, the approach used here costs less

per vaccination than other approaches that intervene less dramatically in the lives of the public (such as approaches using text messages, postcards, or offers of small monetary rewards). It is also novel, in my experience focusing on the USA, in that the researchers thought about the culture of the villages and the extent to which information about vaccines might be made more credible and/or accessible by involving local authorities. I would like to see this approach used in the USA and this paper raised new questions for me: who would be the equivalent of the village authorities in a USA suburb? How might their role in regards public health information and/or public-government relations differ from the authorities in rural Sierra Leone?

B. Originality and significance: if not novel, please include reference

The RCT was well-designed and analyzed and the effects were large as well. This paper should be of immediate interest to people across the social, health, and policy sciences and also to those directly involved in efforts to improve public health. The study is novel in that it bundled together social and informational interventions with logistical ones. The social and informational interventions included: first getting buy-in from village leaders; second holding village meetings where, I imagined, the village authorities spoke and/or signaled their support of vaccination; third walking the village --- either seeking small groups or going door-to-door --- to ensure that information and two-way discussion about vaccination could occur outside of the main meetings. The logistical ones included bringing vaccines and medical personnel to rural villages rather than requiring that people in rural villages travel long distances and pay large costs to get a vaccine.

This paper is also significant for the collaboration between a governmental organization (the Ministry of Health of Sierra Leone), an NGO and social scientists: that is, the intervention used here is of more direct policy relevance both within and outside of Sierra Leone because, I imagine, because policy makers collaborated in the design and implementation. This further increases the importance of these findings and makes me believe that if this approach inspires other governments to use this approach elsewhere in other parts of Sierra Leone or elsewhere, we should see increases in vaccination as a result.

This paper should form a key part of the evidence-base that governments should use to design current and future pandemic responses and vaccination

campaigns. I

think that the results here should convince policy makers to adopt this approach

--- including a randomized evaluation component and formative research to establish who counts as a "village authority" in contexts that are not the same

as rural Sierra Leone. That is, the study is sufficiently rigorous and the effects are sufficiently large and the humanitarian impact is so important (where vaccination helps those vaccinated and also those around them and also humanity in general in regards reducing the creation of new variants), that

policy makers should consider this approach. I recommend continuing evaluations of the approach so that the scientific and policy communities can detect any contextual differences that emerge in new countries and places (and historical moments) and inform the informal global learning agenda about vaccination and public health in the context of pandemics/epidemics in rural areas. My recommendation

for continuing to embed evaluation with implementation is not specific to this

case: contexts vary as do implementations, what works in one place and time may surprise us in regards not working in another place and another

time (or working better than expected). By embedding evaluation we can begin to

ask why an intervention worked well in one place, better in another, and worse

in yet another: and this learning about theory (the "why" question) can help us

produce better interventions in the future.

I think that this is an excellent demonstration of the implementation of science. First, it involves stakeholders and authorities who have responsibilities over the population in the design and implementation of the

intervention: researchers are not professionals at providing large scale public

health to a population whereas this is the job of the professionals in the government. The intervention design appeared to me to arise from collaboration

rather than an attempt at focused theory testing by academics (otherwise they

would not have bundled together the set of interventions that they did). Second,

the nature of the intervention suggests the input from both government actors but also social scientists who are thinking deeply about the social and

psychological processes by which people learn and accept (or resist) new information. Second, by committing to a clear and rigorous impact evaluation,

the government-NGO-academic team adds easy to interpret evidence to the broader

scientific community: this allows science to build on these results either to

disentangle the effects of social hierarchy from information (the meetings, the

mobilizers) from simple cost reduction (bringing the vaccine to the village) or to theorize about synergies among authority, information, and availability. Governments and NGOs resist evaluation for fear of a null effect and in so doing prevent future scientists and future policy makers from learning as much as possible from the efforts of the present. This study shows courageous policy makers and NGO leaders, willing to be wrong in the interest of learning and teaching the broader world about how to confront a humanitarian crisis. Even after the passage of the Evidence-Based Policy-making Act in the US federal government US government agencies are just beginning to see the benefits of transparent evaluation as a way to speed learning for themselves and as a way to teach others. And often leaders of NGOs feel as though they must show success to their funders to maintain their flow of funding (where success is a positive causal effect or positive story, not a story about how a good idea didn't seem to work). The Ministry of Health of Sierra and Concern Worldwide showed exceptional courage and foresight in this project: they realized that they would learn to improve their own operations but also they were willing to enable this project in one place and time to contribute to science and to policy makers confronting similar challenges elsewhere and in the future.

C. Data & methodology: validity of approach, quality of data, quality of presentation

The RCT allows us to set aside concerns that the differences in vaccinations between the villages assigned to receive the bundle interventions and those not assigned to receive it can be explained by factors other than the intervention. The approach overall protected the results from such criticisms (for example, ensuring censuses of the villages *before* randomization, randomization at the village level, etc.). I did not see reports about how the survey data and vaccination data were verified in regards errors of recording. I imagine that the authors teams used common practices to ensure that survey enumerators and those recording vaccination data were actually following protocols but these were not reported in the paper as far as I could tell. Although the primary contribution of this paper is to show how a realistic policy intervention can change COVID vaccination rates, the authors dedicate their section 3.2 to explore whether the intervention increased information, trust in the Ministry of Health, or knowledge about COVID and COVID

vaccines.

They used a survey done in a subset of 45 villages for this task. The results

they present show that villagers in both control and treatment villages were

interviewed (section 5.4 says that the responses to these questions came after

to the intervention in the treatment villages and before the intervention occurred in control villages --- under the assumption that no secular change

would occur in both control and treated villages if the treated villages had no

experienced the intervention during the 4 days of the study period). I did not

understand how those 45 villages related to the underlying research design from

the paper. I see that the survey occurred in 104 villages (although I thought

only 100 were in the treatment group). And I understand that an error ("a coding

error") meant that the authors restricted their analysis to 45 villages.

But I

could not tell how many treated versus control villages were used. I also assume

that, perhaps, 22 treated villages were surveyed, I assume that because the analysis uses "randomization fixed effects", the authors included 22 or 23 control villages from the same experimental strata. Is that right?

Basically,

this part of the paper was unclear in regards its research design and especially

the extent to which we should take these comparisons as justified by randomization within strata or not. Also, it would have been nice to see a baseline comparison between the 45 villages used here and those not used just to

contextualize the results: are these the villages with highest baseline knowledge, trust, etc. or lowest or what?

In that section I liked that the authors showed the effects of the intervention

on what ought to be placebo outcomes, like trust in family and friends, trust in

media, and maybe trust in social media. Since this section appeared exploratory,

I would encourage the authors to state this or otherwise to adjust the width of

their confidence intervals for the multiple tests that they imply. The first

option is easiest, encouraging the reader to take the confidence intervals with

a grain of salt.

D. Appropriate use of statistics and treatment of uncertainties

The authors mostly compare average or sums of village level outcomes across the

villages assigned to treatment versus control. This makes sense to me. They do

not specify whether their standard errors arise from the randomization justified

approach (see the Gerber and Green book on Field Experiments, Chapter 3) or the OLS standard errors (which are biased in this case because of almost certain heteroskedasticity caused by the treatment assignment). When reporting individual level analyses they report using "clustered standard errors" but do not report which ones they choose. I am not concerned about this choice with 150 clusters but when they have only 45 clusters their results could change (perhaps only a small amount) whether they used a standard error that corrects for small numbers of clusters. See the US federal government's advice about calculating standard errors from randomized experiments here: In multiple figures the authors add confidence intervals to either counts (in Figure 2) or proportions (other figures) and say "Error bars represent 95% confidence intervals of treatment estimates." I think this is not correct. I think those are intervals on counts or proportions not differences in counts or proportions. In fact, on Figure 2, if those are CIs on the counts in each group, then I would have expected to see a CI on the control group (after all, the control group is a random subset of the pool of villages and so the calculated number of 5 vaccinations on average per village in the control group would vary with different assignments.)

Section 5.6 Statistical Analysis: Do the authors think that treatment effects are correlated with size of village? If so, a robustness check against the biases that can emerge from this kind of relationship might involve some of the techniques described here

Pet peeve: On page 7 it doesn't make sense to use the word "controls" in the context of a randomized study; I would say "covariates" (after all the PIs randomized as a method of control over covariates).

Section 5.7: On visitors from neighboring villages. Were any of these visitors in villages assigned to the control group? Or were these people from villages that were not a part of the experimental pool? If they were from the control group, worth mentioning this. As I understand it the vaccination rates of the control villages were measured *before* the roll out of the intervention in the treated villages, so I think that even if control group villagers visited the treated villages to get vaccinations, the vaccination rate in the control group should not reflect spillover.

Figure A.4: Do these boxes reflect variation within community (such that,

say,
the teams in the community at the top varied greatly in the number of people they vaccinated?)? Were all vaccinations in a community attributable to a team?
Or did many people just come on their own to the vaccination post? If all counts of vaccinations had to be apportioned to a team, were there cooperative or competitive team dynamics at play here? (Either increasing or decreasing the number of people vaccinated in a village as teams worked with or against each other?) This is an aspect of the intervention bundle that I did not appreciate until Figure A.4: that is, in addition to authority buy-in, large meetings, individual contact, and local vaccines and medical personnel, do we also have some kind of interesting mobilizing team dynamic which can also add to (or subtract from) the effectiveness of the intervention? Maybe this just need clarification.

E. Conclusions: robustness, validity, reliability
I feel like the paper so important that the authors should frame the paper not as if some "supply side" approach differs dramatically from some "demand side" approach, but rather than bringing both supply of vaccine and addressing demand through multiple means in person (via village level social hierarchies which people already use to make judgments, via the mobilizers, via surveys asking people to develop opinions and attitudes and articulate them to people representing governments and experts from outside the village) can be cost effective and has great potential as a package of interventions that governments should use. Whether the cost effectiveness of this intervention ought to be compared to those deployed in places with a lot of electricity and cell phones and postal services is an open and perhaps silly question: would these villagers receive SMS messages or emails or postcards or letters at all? Would the cost of using such interventions on these populations far outweigh the costs of the village visits? (i.e. would they require generators, electrification, cell phone coverage, a postal service, roads, etc..)

From my perspective, the authors spend too much time contrasting the policy of solving logistical problems with the policy of solving social and psychological processes behind vaccine hesitancy. This paper does not, as far as I can see, offer a comparison of the two approaches. It provides strong and important evidence that the proposed bundle of interventions works, but it does not make

any formalized comparisons. In fact, as I understand it, the intervention involved a very intensive social intervention: so, the evidence here is not that merely setting up a vaccine distribution center in rural villages has a strong effect, but rather the intervention involving village elders and leaders and then public meetings including those leaders and the national health authorities *and* bringing the vaccines to the villages (plus having "mobilizers" working hand in hand with the nurses at the centers) worked. For example, the authors note that no additional people were vaccinated in 4 assigned-to-treatment villages where village authorities (?all? ?some?) refused to permit the national health service from offering vaccines: did they really refuse an order from the national government? or was this lack of consent to be a part of a randomized study? In any case, this is an intervention aiming at increasing demand for vaccines even as it is an intervention aiming to increase the supply of vaccines. I think that "demand side" versus "supply side" is a red herring for this paper: it could be mentioned in the conclusion in regards what would have happened had medical teams just showed up in a village and announced vaccine availability versus the given intervention bundle. But I'm not sure that this kind of debate is real --- health economics has long known that health preferences of individuals are generated by those who help supply healthcare (in contrast to the idea that preferences are fixed), and behavioral economics/psychology/students of the survey response have also long known that people come up with their attitudes and preferences on the fly. So, I would not undersell the results of this paper by inventing what I think is not a useful dichotomy: this paper must be published and publicized far and wide as an example of an amazing and timely health intervention for people who have been exceedingly difficult to serve.

Table 2: This is a really useful table for those studying vaccination policy and perhaps useful (with caveats) for the cost per vaccine calculations of this paper but I keep thinking that the authors are mis-interpreting the main point of their paper: which is that bringing free vaccines to villages and getting support from village authorities and having large community meetings and sending people to walk the village to talk with people (knocking on doors or holding

small group meetings) all combined to dramatically increase vaccinations in these rural areas. A *bundle intervention* that included a lot of social support and information had a large effect on people who face large logistical (and perhaps social and informational) barriers to get vaccines like the COVID vaccine. Table 2 helps us see how this bundle of interventions compares to others but (1) the costs of the interventions in table 2 are largely costs in the urban and connected Global North and the costs of doing the same interventions in the rural Global South would be quite different, and (2) the costs and logistics of the village visit style intervention in urban Global North would differ from those described by this paper in Sierra Leone (I would love to see such interventions. I kept wondering what the equivalent approach in suburban USA would look like and whether if we could find the right authorities and mobilizers it would make a big difference or not).

On the call for replication: I would have liked to see a sentence or two about how those replications should differ so as to answer questions raised in this current study: is it really important to disentangle the bundle? I'm not sure it is, from the perspective of saving lives, but perhaps there are aspects of other interventions that might teach us more about how to encourage health behavior in relatively isolated places and/or more broadly.

F. Suggested improvements: experiments, data for possible revision
I recommend a re-framing of the paper. This is just re-writing. I would stop making the "demand side" versus "supply side" comparison since the intervention is not really clearly all "supply side" and since the research design does not actually randomize pure provision of vaccine versus the informational, social, aspects of vaccine take up and acceptance. That is, if there is a debate about supply versus demand, this paper is not well situated to address it. If local governments in the USA debate door to door vaccine provision versus SMS messaging, this is only in the context of bringing vaccines that much closer to people, and, in fact, I suspect that US governments are working on both at the same time --- both bringing vaccines as close to people as possible given infra-structure and personnel and trying to increase demand for vaccines. More importantly, I think that debate is a distraction from the goal of bringing vaccines and health to rural villagers in Africa. And I'm 100% compelled that the difficulty

of getting a vaccine deters most of these villagers from getting the vaccine.

And I would have liked to see some of my confusions about the 45 villages and

other small statistical questions cleared up.

G. References: appropriate credit to previous work?

Two more studies for Table 2. Both show small effects.

"The effect of postcard reminders on vaccinations among the elderly: a block-randomized experiment." Chen, Trump, Hall, Le, Behavioral Public Policy, 2020.

"Lessons for Covid-19 Vaccination from Eight Federal Government Direct Communication Evaluations" (a meta-analysis of 8 nudge style vaccine interventions) Kappes et al, Behavioral Science and Policy, 2021.

H. Clarity and context: lucidity of abstract/summary, appropriateness of abstract, introduction and conclusions

I addressed this above. I think that this paper can be a powerful report on an

important research design if the authors re-frame it. I'm eager to have this

paper in the hands of governments, NGOs and anyone else active in vaccinations

and public health.

Referee #2 (Remarks to the Author):

This paper reports the results of a cluster RCT mobile COVID-19 vaccination clinics in rural districts in Sierra Leone. The trial included a nested study in which some households within one of the intervention groups were randomized to home visits to provide vaccine information and encourage them to get vaccinated. The main findings that stand out in the paper are that the mobile clinics increased vaccination coverage in a community by about 20 percentage points (in the control group, vaccination coverage was ~5%) and that the home visits did not affect vaccination uptake. Although the study provides useful information on what happens when mobile vaccination services are offered in places where there was little to no service delivery previously, I have several concerns regarding the novelty of the findings as well as the methods & reporting of results. With regard to the novelty of this work – there are three points that could be taken into account in the framing of the paper:

One, it is well established and widely accepted that community-based delivery of health services in general can help increase access and uptake. The paper makes several statements in the Introduction and Discussion suggesting that the results from their study are novel in ways that should be surprising to researchers & policymakers seeking to increase vaccination rates in low-income countries. In fact, "reaching people where they are" or "bringing services closer to people" are not only commonly stated recommendations in health promotion efforts but also, they are already being practiced in many countries where there is adequate supply of vaccine doses. For high-income countries like the US, see for example the CDC guidelines for Health Departments on ways to increase COVID-19 vaccinations. <https://www.cdc.gov/vaccines/covid-19/health-departments/generate-vaccinations.html> More relevant to this study, see also the WHO Africa Region's Monthly Bulletin (July 2022) on COVID-19 vaccination, which documents examples of many countries in which mobile clinics or community engagement approaches are being used to increase COVID-19 vaccination coverage. This could be noted in the paper and used to reframe the work as a trial that seeks to determine the effectiveness of such mobile service delivery approaches. <https://apps.who.int/iris/bitstream/handle/10665/361671/CV-20220807-eng.pdf?sequence=1>

Two, the cleanest results in the paper are those for the outcome of vaccination rate (# vaccinated

as percent of those enumerated in census). The finding here that about 20-25% of people get vaccinated comes as a surprise – instead of general tone of the paper suggesting that this was a major success, it actually comes as a surprise that mobile service delivery wasn't even more successful. In many settings, one could argue that when vaccines became easily available – i.e. at nearby clinics or at mobile venues – uptake rose very rapidly to comparable percentages of 20-40%. There are various studies of mobile campaigns of other health services which have achieved extremely high uptake in rural communities – see for example the community-based testing strategies that are now widely implemented in many countries in Africa.

<https://www.nature.com/articles/nature16044>

Third, it does not seem reasonable to compare across vaccine promotion studies in the way that the authors do in Table 2. Surely we need to recognize that demand side interventions may be necessary (and will have smaller effects) after the first wave of vaccine adoption takes place. I.e. supply side interventions like mobile service delivery in places where there was no supply previously will pick up the initial 20-40% who are interested in getting vaccinated, and after that one may need other interventions that target demand-side factors. The comparison of the mobile service delivery approach to other vaccination promotion interventions strikes me as apples and oranges – very different settings with very different baseline vaccination rates. For example, in the Campos-Mercade et al. RCT of incentives and behavioral nudges, baseline vaccination rates were 71%! How does it make sense to compare effectiveness of the incentives & nudges in that setting to the effectiveness of a mobile clinic in communities where barely anybody had been vaccinated initially? I would suggest paring back this section quite a bit and focusing on the results of your cluster RCT without too much additional commentary or comparison across studies. That could be a separate paper perhaps.

Major comments

1. It seems odd for a primary outcome to be measured at baseline in the control group and at endline in the intervention groups. In a rigorous RCT, one would have a follow-up data from the control group as well. I recognize you are using baseline data since only 5-6 days elapsed, but still...it would almost have been better to collect data in control clusters at endline rather than baseline in this case. I think the comparison to baseline data from the control group is problematic.
2. Rather than leading with Figure 2 in the results section, I would suggest presenting Figure 3 first since it is the primary outcome. I would also suggest reporting the control group vaccination rate in this figure
3. For measuring the primary outcome of vaccination uptake, should we be concerned about measurement error? I.e. the possibility that some individuals who were enumerated did get vaccinated but did not have their vaccination card with them?
4. Please report the power calculations and sample size determinations that were done at the outset of the study. This does not appear to be included in the paper.
5. The effectiveness of the home visits would seem to depend quite a bit on implementation factors – i.e. fidelity to the intervention and time spent at each house that was selected – as well as the quality of the information & interactions. How was the curriculum developed? Did the team collect information on implementation factors?
6. Did the study measure distances from households to the vaccination site? If yes, it would be useful to report average distances and examine whether distance is associated with uptake. If no, then it would be helpful to give an indication of approximate travels times/costs associated with coming for a vaccine.
7. Doesn't the possibility of spillover effects across households reduce the ability to detect an effect of the home visits (Table 1)? I would suggest discussing this possibility and also adjusting for it if possible (possible with spatial adjustment if there is information on proximity to treated households).
 - a. Minor point: the term "door knock" as used in the paper and in the tables is rather awkward and casual. Consider replacing it with terms like "Home visits" or "interpersonal communication"
8. Regarding the authors' recommendation that community meetings be widely implemented, how practical is this and shouldn't the CEA analysis adopt a household perspective and account for costs incurred by households as well any costs that would be incurred in order to increase meeting

attendance?

9. What COVID-19 vaccine did the NGO offer in the communities? Important information that doesn't seem to be mentioned anywhere.
10. What was the search strategy used to identify "recent studies" that sought to increase vaccine uptake? This does not appear to be described in the paper and I think it is important to explain the databases and search terms that were used. Also, I suggest that Table 2 have a different title other than "comparing vaccination rates".
11. Were the sub-group analyses pre-specified? I'd recommend indicating this in the methods including the exact sub-groups that were pre-specified.
12. Community and home-based service delivery have been compared to facility-based service delivery in many contexts and it is widely accepted that bringing services closer to people increases access and uptake. Consider citing other studies that have shown such results. Here is a review article in Nature that establishes this for HIV services
<https://www.nature.com/articles/nature16044>
13. Cite other studies have done similar comparisons of mobile vs clinic based vaccination – e.g. this one from Switzerland: https://papers.ssrn.com/sol3/papers.cfm?abstract_id=4166827
14. I am surprised that aside from the director of the EPI program in Sierra Leone, there were no local academic researchers or senior research staff involved as co-authors.
15. In line with my earlier points, abstract could revised to report the main methods & results. These sentences seem better suited for discussion/interpretation – the second sentence doesn't seem factually true given that the study did not achieve anywhere close to universal coverage. "This has led to an explosion of scholarly and policy interest on vaccine hesitancy and ways to overcome it. We argue instead that the limiting factor for many rural Africans is lack of access to convenient vaccine services, and implemented an intervention designed to overcome the logistical challenges of last mile service delivery"

Minor comments

1. It would be useful to report the exact % vaccinated in each to the three clusters in addition to the regression results. Currently it is not easy to know the exact % of population vaccinated in each of the two intervention groups (for the control group you report the % in the regression results in Fig 2 and one has to use the regression estimates in Fig 3 to work this out). I would recommend replacing Figure 3 with a simple table that shows vaccination rates in each study group (including control) and the effect size of the interventions as betas.
2. There are several places with text that is unnecessary or somewhat removed from the main focus of the paper. For example, there is a discussion on page 18 about whether vaccination is necessary. This strikes me as besides the main goal of this paper, which is a scientific study of whether the mobile clinic strategy increases vaccination coverage and the corresponding cost-effectiveness of that strategy. I suggest reducing the length of this already long paper by focusing on the main analytical components of the paper.
3. There is repetition of text in various places. Reducing such repetition would make the paper much easier to read. Examples include:
 - a. Pubmed searches about demand side vs supply side approaches (repeated in Introduction and Discussion)
 - b. In the Results section, the vaccination rate is defined multiple times – with the point about the census & denominator being mentioned on page 6 and 7. I would suggest clearly defining the vaccination rate in Section 2 and then reporting the results only in the Results section.
4. Reduce the number of digits in the tables, especially those in the Appendix
5. Remove statistical tests of differences between study groups from Table 1. These don't seem necessary to me given that this was an RCT
6. Consider using past tense when describing the study results. Currently there is a mix of present and past tense being used (with present tense more commonly used) – uniformity in the language would make the paper easier to read.
7. Consider using "increased to" instead of terms like "vaccination rate jumps to.."

Referee #3 (Remarks to the Author):

This paper offers a compelling argument to dispel the vaccine hesitancy myth drawing on the results of the Expanded Programme on Immunization in Sierra Leone. The 20% increase in vaccine up take supports the primary argument presented in the paper demonstrating that people living in rural Sierra Leone were not deeply hesitant and that providing access to vaccines was an effective way to increase vaccination coverage.

The article's reflections on academic literature and funding priorities that reinforce the perception of vaccine hesitancy across the African continent, is profoundly important in attempting to reshape the narrative on vaccine access. The partnership with the Sierra Leone Ministry of Health, further emphasizes the critical role of working alongside governments to ensure access to vaccines and healthcare more generally.

Very importantly, the article demonstrates the value of working with community leaders in rural areas to ensure vaccine uptake, reinforcing the need to plan interventions with communities.

Based on this assessment, I recommend the paper for publication.

Nature MS #2022-09-14388

“Solving Last-Mile Delivery Challenges is Critical to Increase COVID-19 Vaccine Uptake: A Cluster Randomized Controlled Trial”

Response to editor and reviewers

We thank the editor and three reviewers for the extremely thoughtful comments and helpful suggestions. We appreciate the opportunity to revise our manuscript. By responding to the excellent comments raised, we believe that the enclosed revision is much improved relative to the initial submission. In what follows, we reproduce all comments (indented and in *italics*), and then respond to each comment directly in blue.

Dear Editor,

We are grateful to you and the referees for the thorough and constructive assessment of our paper. The revised paper incorporates all substantive suggestions from referees. We believe we were able to address the concerns raised by R2, while highlighting the strengths of the paper that R1 and R3 appreciated.

As we understand from both your cover letter and the comments sprinkled throughout the R2 referee report, the main concern R2 has is about novelty relative to our existing knowledge base. Versions of the same comment show up in the following forms (and we are paraphrasing here):

“WHO and CDC guidelines state that...” or “a working paper in Switzerland has shown that...” or “it is widely accepted that...” - “mobile service delivery that brings services closer to the people increases uptake.”

R2 is correct that our study and the Swiss working paper have one thing in common - they both demonstrate that access is important. The Swiss study demonstrates that making access convenient in a high-income country in the earliest phase of vaccine distribution (August 2021 in Switzerland) speeds up vaccine uptake. But there are some fundamental ways in which our study, its purpose, its context, and its policy consequences differ from the type of effort embodied in the Swiss working paper.

Unlike Switzerland and other high-income countries, where everyone who wanted a vaccine could easily access one by late 2021, the vaccine still has not reached the majority of the rural Africans. The logistical challenge of last-mile delivery in remote rural parts of Sierra Leone (and elsewhere in Africa) is fundamentally different from efforts in Switzerland, or any other high-income nation.

It is important to *demonstrate* that this challenging task can be completed cost-effectively in remote parts of Sierra Leone, for reasons beyond the academic goal of rigorously establishing that access is important. Establishing that “proof-of-concept” in the most remote, logistically challenging context allows us to mobilize partners and build the necessary coalitions of donors, health ministries, NGO partners, and researchers to replicate and scale that same implementation strategy in other remote regions of Sierra Leone and in other low-income

countries. This is why we are eager to publish in a prominent outlet such as *Nature*, because the results need to get noticed, and once it does, it will mobilize the resources, attention, and effort of relevant stakeholders to solve this problem on a larger scale. Remarkably, the pre-print uploaded by *Nature* to an archive at the time of our original submission has already opened up those paths. It has led to partnerships with UNICEF's global team on immunization and two different regional teams on how our results could inform their strategies for reaching "zero dose" populations globally. Gavi has expressed interest in adopting the same approach to reach "zero dose" populations. The Social Science Research Council (SSRC), which recently allocated \$20 million in funding for vaccine uptake research, has highlighted this preprint (see screenshot of their tweets at the end of this letter), and invited us to apply for funding for follow-up research.

This is also why in our revised paper we conduct a thorough systematic review (following R1's advice) to properly benchmark our effect size and cost-effectiveness against other vaccination efforts around the world, to make the case that this is a cost-effective exercise that should be scaled in other low-income settings.

These partnerships have formed because the potential applications of this proof of concept go beyond COVID vaccines. As we discuss towards the top of page 2 of our Introduction, the same strategy can be applied to deliver bundles of maternal and child health interventions - including routine immunizations - in those same communities. In fact, bundling multiple interventions would make the effort even more cost-effective than our RCT with COVID vaccines described in this manuscript. That is precisely the next step of our program, leveraging the coalition we have formed through this project. We are now preparing for the implementation of this "bundled approach" to routine immunization with the Sierra Leone Ministry of Health.

All of this is important because routine immunization rates in Sierra Leone and neighboring countries in West Africa are low, and infant and child mortality rates too high, even relative to, say, Bangladesh. So the broader applicability of our study and its potential to trigger large-scale action are fundamentally different from a study conducted in Switzerland. And demonstrating in the real world that it can be done cost-effectively serves a fundamentally different purpose than writing down in a WHO or CDC guideline document that this is conceptually a good idea to try out, which is the other type of knowledge cited by R2.

We have rewritten and reframed the introduction entirely to eliminate any claims of conceptual novelty. We have instead adopted the language of "proof of concept" to explain the value of this research, because the goal is to go beyond the academic exercise and use the results as a catalyst for change. We are forming partnerships to turn these research findings into actionable steps that address real-world challenges at scale, as explained in this letter and in the revised manuscript.

Positioning relative to the Academic Literature

The final question is whether our work is novel from an academic standpoint, or if there are other papers in the literature like the Swiss working paper. The authors of the Swiss working

paper state in their paper: “Despite the popularity of MVUs [mobile vaccination units], so far, no randomized controlled trial (RCT) has been reported to test their effectiveness”. In the process of conducting our systematic literature review via PubMed, we found that this is true.

In the specific context of a pandemic like COVID-19 - which placed a sudden burden on medical teams globally, leading to an urgent scramble for policy answers to increase vaccine uptake - we felt that the eminently sensible case for improving vaccine accessibility was undermined by the paltry academic attention to supply-side solutions, relative to the overwhelming literature on vaccine hesitancy. This is consistent with the results of our systematic literature review, which vindicate Reviewer 1’s comments about the value of this exercise (see **sections 4.1, 4.2, Figures 6 and 7, and the large Supplementary Table A10**). Scientific evidence for mobile vaccination units (and the necessary proof of concept) are indeed lacking.

Our new systematic review of Randomized Controlled Trials on Vaccine uptake published between 2000 and 2023 produces over 3600 hits. We group the studies into 5 themes (incentives, communication, education, etc), and the papers that come closest to our approach are grouped under "Community Actions". This is the smallest group representing <5% of the RCT literature on vaccine uptake, and it includes 12 interventions in 7 studies. In the most literal sense, we did not find *any* published paper that claimed to test the marginal effect of mobile vaccination units on vaccine uptake relative to a pure control. However, as highlighted by Reviewer 2, a limited number of studies (three, as could be identified from our literature review) have attempted RCTs with the ostensible objective of supplying vaccines to remote or inaccessible populations in low-income contexts. These studies, all of which were conducted prior to the COVID pandemic, suffer from one or more limitations: two of them do not report treatment effects relative to a *pure* control group, while the third study, which does report results relative to a pure control, uses a year-long intervention.

- Sengupta et al. (2017) worked with "government Medical Officers responsible for [study areas] as well as the community...to establish a government funded outreach vaccination programme in settings provided by the community". Effect size is 17.5pp for VPD vaccines
- Ateudjieu et al. (2022): use monthly data recorded by community volunteers in the Fouban region of Cameroon to coordinate visits from a health center immunization team for catch-up immunization sessions. Finds a zero effect.
- Habib et al. (2017): A community-based three-arm cluster randomised trial in three districts of Pakistan at high risk of polio. The two interventions both included community outreach and mobilisation, using an enhanced communication package and provision of short-term preventive maternal and child health services and routine immunisation. Finds a 7 percentage point effect, and 9 percentage points for oral polio.

Summary

If what we show was all well-known and easily do-able, then vaccination rates in rural African communities would not be in single digits even 2 years after COVID vaccines arrived on the market. The concept we demonstrate via this RCT can be applied more broadly to deliver other important maternal and child health interventions including routine immunizations in remote,

rural areas. We now make that case in a revised introduction and **Section 4.3 “Policy Implications”**. The response to the pre-print uploaded by *Nature* has already moved us in that direction. We hope and expect that a published paper in *Nature* will be transformative because it will lead to scale-up and replication of this exercise.

Our revised manuscript is also responsive to all the detailed comments from all referees. For example, we have re-organized the presentation of the results prioritizing effects on the *vaccination rate* (request from R2), and provide additional tests and statistics in **Supplementary Tables A2, A3, A4, A5, A9** (requests from R1).

We have rewritten the introduction and reframed the paper entirely, to move away from the “supply vs demand” framing which R1 and R2 rightly advised us to move away from.

We should note here that the treatment effects in **Figure 2** and **Figure 3** are slightly larger than presented before. This is due to a small coding error and changes in the eligible population variable. The coding error related to a manual fix of the age variable in the data. We had used the exit survey of the vaccination drive to update respondent characteristics (predominantly age status). Some respondents were previously miscoded as being below 18 and thus not eligible for a vaccine. However, on inspection of the exit survey they were above 18. Because we did not have an equivalent additional check on age in control villages, this correction created a (downward) bias in our estimates. This has now been fixed resulting in slightly larger estimates.

Response Memo: Reviewer 1

*Review of "Solving Last-Mile Delivery Challenges is Critical to Increase COVID-19 Vaccine Uptake: A Cluster Randomized Controlled Trial"
Nature 2022-09-14388*

A. Summary of the key results

This paper shows that bringing vaccines and community meetings and getting buy-in from village authorities and local mobilization to people living in the most remote areas in Sierra Leone has a huge effect on vaccination rates there. Compared to approaches to encouraging vaccination common outside the Global South, the approach used here costs less per vaccination than other approaches that intervene less dramatically in the lives of the public (such as approaches using text messages, postcards, or offers of small monetary rewards).

B. Originality and significance: if not novel, please include reference

The RCT was rigorous and large scale and the effects were large as well. This paper should be of immediate interest to many people across the social, health, and policy sciences and also to those directly involved efforts to improve public health. The public policy studies is novel in that it bundled together social and informational interventions (first getting buy-in from village leaders; second holding village meetings where, I imagined, the village authorities spoke and/or signaled their support of vaccination; third walking the village (either seeking small groups or going door-to-door) to ensure that information and two-way discussion about vaccination could occur outside of the main meetings with logistical ones (bringing vaccines and medical personnel to rural villages rather than requiring that people in rural villages travel long distances and pay large costs to get a vaccine). This paper is also significant for the collaboration between a governmental organization (the Ministry of Health of Sierra Leone) and social scientists: that is, the intervention used here is of direct policy relevance because, I imagine, it was designed in collaboration with policy makers and was also implemented in close collaboration with those policy makers. This further increases the importance of these findings and makes me believe that if this approach inspires replications elsewhere in other parts of Sierra Leone or elsewhere, we should see increases in vaccination as a result.

This paper should forms a key part of the evidence-base that governments should use to design current and future pandemic responses and vaccination campaigns.

Thank you for your very constructive and helpful comments. We have taken into account all of your excellent suggestions, and provide point-by-point responses below to make it easy to track the edits.

C. Data & methodology: validity of approach, quality of data, quality of presentation

*The RCT allows us to set aside concerns that the differences in vaccinations between the villages receiving the bundle interventions and those not receiving it can be explained by factors other than the intervention. The approach overall protected the results from such criticisms (for example, ensuring censuses of the villages *before* randomization, randomization at the village level, etc.). I did not see reports about how the survey data and vaccination data were verified in regards errors of recording. I imagine that the authors teams used common practices to ensure that survey enumerators and those recording vaccination data were actually following protocols but these were not reported in the paper as far as I could tell.*

Thank you for raising this point. Both the team of enumerators in charge of the survey, and the team of social mobilizers and vaccinators from the Ministry of Health and Sanitation received extensive training on implementation protocols and record keeping prior to any field activity. Only those individuals that were considered as proficient after examination were retained for the data collection and implementation of this study. We have added a note to this effect in the manuscript in **Section 5.5 on page 22**.

Baseline vaccination rates in all villages were measured during the village census and verified by asking respondents to provide evidence of vaccination by presenting their vaccination cards. Vaccination cards were inspected and the vaccination status and the number of vaccine doses received was recorded. It is worth noting that it is common practice in Sierra Leone to keep vaccination and for example voter ID cards safe and dry. The verified vaccination rate is the standard measure proposed by the Ministry of Health. Unverified vaccination rates could produce measurement error. Unverified vaccination rates could produce measurement error, as self-reports can be influenced by recall and social desirability bias. At baseline 1,116 reported having had a COVID-19 vaccine for which we could verify the vaccine card. For an additional 1,555 (spread equally across treatment arms) we could not verify the vaccine card. These have been left out of the analysis (such as the count and rate values that appear in **Figure 2 and 3**).

To measure vaccine uptake due to our intervention, in treated communities, our data collection teams worked in close collaboration with the vaccine teams and conducted an exit survey where they again followed the same protocol and inspected the vaccination card to record the vaccination status, only 6 people did not consent to be surveyed.

For all our surveys (census, baseline, exit survey, and endline) were recorded unique identifiers for all respondents (code, first, middle, surname, age, gender, phone number, head of their household, etc.). This allowed enumerators to correctly identify all respondents and (should the

enumerators make recording mistakes) the data analyst to spot and correct possible mismatches. Vaccination cards were photographed to allow team supervisors to inspect whether the information in our data was correct. We recorded very few data entry mistakes.

*Although the primary contribution of this paper is to show how a realistic policy intervention can change COVID vaccination rates, the authors dedicate their **section 3.2** to explore whether the intervention increased information, trust in the Ministry of Health, or knowledge about COVID and COVID vaccines. They used a survey done in a subset of 45 villages for this task. The results they present show that villagers in both control and treatment villages were interviewed (section 5.4 says that the responses to these questions came after to the intervention in the treatment villages and before the intervention occurred in control villages --- under the assumption that no secular change would occur in both control and treated villages if the treated villages had no experienced the intervention during the 4 days of the study period. I did not understand how those 45 villages related to the underlying research design from the paper. I see that the survey occurred in 104 villages (although I thought only 100 were in the treatment group). And I understand that an error ("a coding error") meant that the authors restricted their analysis to 45 villages. But I could not tell how many treated versus control villages were used. I also assume that, perhaps, 22 treated villages were surveyed, I assume that because the analysis uses "randomization fixed effects", the authors included 22 or 23 control villages from the same experimental strata. Is that right? Basically, this part of the paper was unclear in regards its research design and especially the extent to which we should take these comparisons as justified by randomization within strata or not. Also, it would have been nice to see a baseline comparison between the 45 villages used here and those not used just to contextualize the results: are these the villages with highest baseline knowledge, trust, etc. or lowest or what*

Thank you for your comment. Indeed, our data on these secondary outcomes were collected in a subset of both treatment and control villages. In the paper, we report on 45 villages which are balanced over the three treatment arms, ie 15 door-to-door and 15 small group and 15 control villages. There was nothing systematic about this choice - we just realized midway through that this section of the survey was missing in the early rounds of data collection.

We actually collected these data in more villages - 50 control villages, 30 Door to Door treatment arm, and 25 Small Group treatment arm - for a total of 105 villages. However, we only observe all three treatment arms in 15 blocks, i.e. 45 villages. We therefore restrict our reporting in the paper to those 45 villages to be consistent with the way we analyze the data in all other parts of the paper.

In the revised manuscript we have expanded the text describing this subsample to make the logic behind this choice clearer in the **Methods section 5.4, p.21**. We have also included an additional **Table A9** in the Supplementary Materials, where we compare characteristics of the

subsample to the full sample. The subsample is on average very similar to the overall sample, an overall F-test does not reject the equality of means, p -value = 0.662.

We will take this opportunity to also explain why we visited the control villages only once at baseline, and its implications. We agree that ideally we would capture two measurements (baseline and endline) in each village. Our team considered the trade off, and the decision for one measurement was largely informed by the following logic. It is highly unlikely that these remote places visited by our vaccination teams would have been visited by other health personnel from MoHS in the ~5 day period between baseline and midline, or that a large number of people would have incurred the cost of visiting the CHC for receiving a vaccine. To substantiate this argument, we can use the fact that our baseline survey was conducted over a few weeks across communities, so we inspect the temporal trends in the data. We test the potential natural increase in vaccination rates over time by assessing the change in vaccination rates over the course of the rollout of the baseline survey. In the figure below we present the baseline vaccination rate (y-axis) over time including a regression line with 95% confidence intervals. There is no apparent trend, suggesting that vaccination rates did not increase over time and reducing worries that our choice to not revisit control villages impacts on the endline results. We now acknowledge this concern and describe this test in the subsection 5.4 “Data Collection” in our “**Methods**” section.

We verified that there were no vaccination drives being conducted in this period. The costs of revisiting communities in these remote locations are high (note that in our CEA analysis the cost of transport are the largest line item and about 33% of the total implementation costs, see **Table A8**), so we opted for a single visit to stay within budget and use that money to increase our sample size instead.

In that section I liked that the authors showed the effects of the intervention on what ought to be placebo outcomes, like trust in family and friends, trust in media, and maybe trust in social media. Since this section appeared exploratory, I would encourage the authors to state this or otherwise to adjust the width of their confidence intervals for the multiple tests that they imply. The first option is easiest, encouraging the reader to take the confidence intervals with a grain of salt.

Many thanks for this suggestion, we have updated the text describing the results of the **Figure 4 (on p11)** to make this point explicitly, and have added FDR q-values to the regression output in **Table A4** and **Table A5**.

D. Appropriate use of statistics and treatment of uncertainties

The authors mostly compare average or sums of village level outcomes across the villages assigned to treatment versus control. This makes sense to me. They do not specify whether their standard errors arise from the randomization-justified approach (see the Gerber and Green book on Field Experiments, Chapter 3) or the OLS standard errors (which are biased in this case because of almost certain heteroskedasticity caused by the treatment assignment). When reporting individual level analyses they report using "clustered standard errors" but do not report which ones they choose. I am not concerned about this choice with 150 clusters but when they have only 45 clusters their results could change (perhaps only a small amount) whether they used a standard error that corrects for small numbers of clusters. See the US federal government's advice about calculating standard errors from randomized experiments here:

<<https://oes.gsa.gov/assets/files/calculating-standard-errors-guidance.pdf>>

Thank you for raising this point. For our main results on the vaccine count and rate we estimate the Intent To Treat effects with fixed effects and heteroskedasticity-robust standard errors using a generalization of a linear estimator to account for high dimensional fixed effects based on Correia (2014, see <http://scoreia.com/software/reghdfe/index.html>). We have updated our text in the **Methods** section to make this clear.

For robustness, we also re-estimate our main effects and the smaller subsample using bootstrapped standard errors (following Cameron, Gelbach & Miller RevEcStat 2008). We report the bootstrapped p-value, in **Tables A2, A3, A4, and A5**, resulting from a wild bootstrap test of pooled treatment == 0 using 999 repetitions. The overall interpretation of the results in **Tables A4 and A5** does not meaningfully change.

In multiple figures the authors add confidence intervals to either counts (in Figure 2) or proportions (other figures) and say "Error bars represent 95%

confidence intervals of treatment estimates." I think this is not correct. I think those are intervals on counts or proportions not differences in counts or proportions. In fact, on Figure 2, if those are CIs on the counts in each group, then I would have expected to see a CI on the control group (after all, the control group is a random subset of the pool of villages and so the calculated number of 5 vaccinations on average per village in the control group would vary with different assignments.)

Thank you for raising this point. We have corrected Figures 1 and 2 and include the CI's throughout. We realized the presentation of the main results was confusing.

We also followed your suggestions as well as R2's suggestion and have reorganized our presentation of the results. In **Section 3**, we now first present the *vaccination rate* pooled across treatment arms (**Figure 1**, ie for the village population as recorded in our census), and include the CI for the baseline and endline rate in pooled treatment and control villages.

In the next section present the results for the *vaccination count* (**Figure 2**, ie for all those that were vaccinated regardless of residency) and again include the CIs for the baseline count, and endline count for each subgroup.

Section 5.6 Statistical Analysis: Do the authors think that treatment effects are correlated with size of village? If so, a robustness check against the biases that can emerge from this kind of relationship might involve some of the techniques described here

<<https://declaredesign.org/blog/bias-cluster-randomized-trials.html>>

Thank you for this comment. In our randomization process we did not block on village size (which would be a design way avoiding potential bias in the OLS estimates). As mentioned in the **Methods section on page 20**, we did block on other variables (i) the baseline share of the population that was immunized, (ii) the age of the population, (iii) literacy levels, and (iv) the distance from the closest clinic. Note that village size is balanced, see **Table A1**. Below, we re-estimate our model for the vaccination rate adding dummies indicating the quantiles of village size as covariates. The treatment estimate does not change and the dummies each enter with with small coefficients and are not statistically significant. See the regression output here:

```
. reghdfe vaccinated_endline anytreat vaccinated_baseline i.villsize_q, absorb(grpID)
(MWFE estimator converged in 1 iterations)
```

```

HDFE Linear regression      Number of obs =      150
Absorbing 1 HDFE group     F(   5,   95) =     49.27
                             Prob > F       =     0.0000
                             R-squared       =     0.7793
                             Adj R-squared  =     0.6538
                             Within R-sq.   =     0.7217
                             Root MSE    =     0.1172

```

vaccinated_endline	Coef.	Std. Err.	t	P> t	[95% Conf. Interval]	
anytreat	.2602877	.021169	12.30	0.000	.2182618	.3023135
vaccinated_baseline	.612813	.0960405	6.38	0.000	.4221485	.8034774
villsize_q						
2	.0306464	.03319	0.92	0.358	-.0352441	.0965368
3	.0181358	.0319035	0.57	0.571	-.0452006	.0814722
4	-.0474851	.0326071	-1.46	0.149	-.1122185	.0172483
_cons	.0197636	.0273868	0.72	0.472	-.0346061	.0741333

Also when we interact treatment with these quantile dummies, none of the coefficients are significant, see our regression output copied below.

```
. reghdfe vaccinated_endline anytreat vaccinated_baseline anytreat##villsize_q, absorb(grpID)
(MWFE estimator converged in 1 iterations)
note: 1.anytreat omitted because of collinearity
```

```

HDFE Linear regression      Number of obs =      150
Absorbing 1 HDFE group     F(   8,   92) =     32.21
                             Prob > F       =     0.0000
                             R-squared       =     0.7914
                             Adj R-squared  =     0.6621
                             Within R-sq.   =     0.7369
                             Root MSE    =     0.1158

```

vaccinated_endline	Coef.	Std. Err.	t	P> t	[95% Conf. Interval]	
anytreat	.2413544	.049535	4.87	0.000	.1429737	.3397351
vaccinated_baseline	.6022445	.0964393	6.24	0.000	.4107077	.7937813
1.anytreat	0	(omitted)				
villsize_q						
2	-.0369498	.055849	-0.66	0.510	-.1478707	.0739711
3	.0161486	.0612253	0.26	0.793	-.1054501	.1377474
4	-.0201	.0606023	-0.33	0.741	-.1404615	.1002614
anytreat##villsize_q						
1 2	.1130712	.0718974	1.57	0.119	-.0297232	.2558655
1 3	-.0010341	.073785	-0.01	0.989	-.1475776	.1455093
1 4	-.0506407	.0711964	-0.71	0.479	-.192043	.0907616
_cons	.0378064	.0414892	0.91	0.365	-.0445947	.1202075

Pet peeve: On page 7 it doesn't make sense to use the word "controls" in the context of a randomized study; I would say "covariates" (after all the PIs randomized as a method of control over covariates).

Corrected, we replaced “controls” with “covariates”. As mentioned, we have updated the presentation of the results (see above). In the regression tables, **Tables A2 and A3** we now present the ITT estimates with and without covariates for the baseline vaccination rate, and covariates that were imbalanced at baseline; gender, age, education, owning land.

*Section 5.7: On visitors from neighboring villages. Were any of these visitors in villages assigned to the control group? Or were these people from villages that were not a part of the experimental pool? If they were from the control group, worth mentioning this. As I understand it the vaccination rates of the control villages were measured *before* the roll out of the intervention in the treated villages, so I think that even if control group villagers visited the treated villages to get vaccinations, the vaccination rate in the control group should not reflect spillover.*

During the exit survey, enumerators asked respondents who reported to have come from a different village about the District and Village name. We then matched these names back to our list of control villages. Using a hard match on District names and then a Levenshtein distance metric to match village names (this is a text similarity measure that compares two words and returns a numeric value representing the distance between them). Allowing for a string distance of 2, we find only 8 matches. Using a more conservative cut-off of 1, find no overlap whatsoever. In summary, the spillover benefits largely accrued to others who were not part of the experimental pool.

The minimum as the crow flies distance between project treatment and control villages was 8.5 miles, so at least a 2-3 hour walk. That makes it highly unlikely that any other experimental project village was indirectly affected by the treatment. We now mention this in Section 5.4 on **page 22**.

Figure A.4: Do these boxes reflect variation within community (such that, say, the teams in the community at the top varied greatly in the number of people they vaccinated)? Were all vaccinations in a community attributable to a team? Or did many people just come on their own to the vaccination post? If all counts of vaccinations had to be apportioned to a team, were there cooperative or competitive team dynamics at play here? (Either increasing or decreasing the number of people vaccinated in a village as teams worked with or against each other?) This is an aspect of the intervention bundle that I did not appreciate until Figure A.4: that is, in addition to authority buy-in, large meetings, individual contact, and local vaccines and medical personnel, do we also have some kind of interesting mobilizing team dynamic which can also add to (or subtract from) the effectiveness of the intervention? Maybe this just need

clarification.

Vaccination uptake in a village reflects both village characteristics, as well as vaccine team effort. We did not randomly assign vaccine teams to (strings of) villages as this was logistically impossible, and also undesirable from the Ministry's perspective because they wanted to guarantee team ability and composition matched the type of village (i.e., the more physically fit ministry staff were sent to the most remote places). Teams did not receive any performance-based incentives: all teams were remunerated per day of work irrespective of the number of people vaccinated. Teams could only monitor their own performance (i.e. they had no information about the performance of other teams). There are no specific team dynamics that we could measure, other than team fixed effects we present in **Figure A4**. Each village was visited by one vaccine team and in our experience, performance is more likely a function of the intrinsic motivations of team members (which can naturally vary across individuals) and less about village characteristics. It is however impossible to fully separate the two effects empirically, given our implementation strategy. We thus only speculate that providing good performance incentives to teams could improve the cost-effectiveness of this effort.

E. Conclusions: robustness, validity, reliability

I feel like the paper so important that the authors should frame the paper not as if some "supply side" approach differs dramatically from some "demand side" approach, but rather than bringing both supply of vaccine and addressing demand through multiple means in person (via village level social hierarchies which people already use to make judgments, via the mobilizers, via surveys asking people to develop opinions and attitudes and articulate them to people representing governments and experts from outside the village) can be cost effective and has great potential as a package of interventions that governments should use. Whether the cost effectiveness of this intervention ought to be compared to those deployed in places with a lot of electricity and cell phones and postal services is an open and perhaps silly question: would these villagers receive SMS messages or emails or postcards or letters at all? Would the cost of using such interventions on these populations far outweigh the costs of the village visits? (i.e. would they require generators, electrification, cell phone coverage, a postal service, roads, etc..)

From my perspective, the authors spend too much time contrasting the policy of solving logistical problems with the policy of solving social and psychological processes behind vaccine hesitancy. This paper does not, as far as I can see, offer a comparison of the two approaches. It provides strong and important evidence that the proposed bundle of interventions works, but it does not make any formalized comparisons. In fact, as I understand it, the intervention involved a very intensive social intervention: so, the evidence here is not that merely setting up a vaccine distribution center in rural villages has a strong

*effect, but rather the intervention involving village elders and leaders and then public meetings including those leaders and the national health authorities *and* bringing the vaccines to the villages (plus having "mobilizers" working hand in hand with the nurses at the centers) worked. For example, the authors note that no additional people were vaccinated in 4 assigned-to-treatment villages where village authorities (?all? ?some?) refused to permit the national health service from offering vaccines: did they really refuse an order from the national government? or was this lack of consent to be a part of a randomized study? In any case, this is an intervention aiming at increasing demand for vaccines even as it is an intervention aiming to increase the supply of vaccines. I think that "demand side" versus "supply side" is a red herring for this paper: it could be mentioned in the conclusion in regards what would have happened had medical teams just showed up in a village and announced vaccine availability versus the given intervention bundle. But I'm not sure that this kind of debate is real --- health economics has long known that health preferences of individuals are generated by those who help supply healthcare (in contrast to the idea that preferences are fixed), and behavioral economics/psychology/students of the survey response have also long known that people come up with their attitudes and preferences on the fly.*

So, I would not undersell the results of this paper by inventing what I think is not a useful dichotomy: this paper must be published and publicized far and wide as an example of an amazing and timely health intervention for people who have been exceedingly difficult to serve.

Thank you for this very insightful comment. We agree with you 100%, and we have entirely rewritten the introduction and reframed the paper accordingly. We have removed the demand versus supply side framing. In the revised introduction, we only write a couple of sentences about what our RCT establishes: that access is a critical factor for vaccine uptake in remote areas of low-income countries, and more attention needs to be paid to deficiencies in access.

Table 2: This is a really useful table for those studying vaccination policy and perhaps useful (with caveats) for the cost per vaccine calculations of this paper ...

Many thanks. We have now completely updated the former **Table 2**. We have updated our literature overview and now present the results of a systematic review of RCTs that include approaches to increase vaccination uptake. We included a new **sub-section 5.7 in Methods** section that describes our search procedure.

We searched PubMed of papers published between Jan 1, 2000 and Jan 7, 2023 using the search terms "(vaccin*[Title/Abstract] OR immun*[Title/Abstract]) AND \textit{additional search term}[Title/Abstract]) AND (Randomized Controlled Trial[Publication Type])", with additional search terms: ``access"; ``community-based"; ``cost effect*"; ``demand"; ``hesitant";

``incentive*"; ``intervention*"; ``mobile"; ``nudge*"; ``rural"; and ``supply". These searches yielded 3,615 unique articles. We screened out articles that are not related to vaccine uptake or that did not rely on an RCT, reducing the sample to 141 articles. We appended a further 20 relevant studies that were identified by snowballing and rejected 17 papers that did not have a control group; did not report the percentage-point change in vaccine uptake; and did not include a test statistic. The final list of 144 articles comprise 234 unique interventions for which we can report a percentage-point change relative to a control group.

We then grouped these studies as information, nudges, community engagement, social signaling, non-financial and financial incentives. The mean effect sizes per group is included in the new **Figure 6**. The full list of papers is now in the Supplementary Materials, **Table A10**.

Of the final list, 33 interventions (14.10%) report information about the cost of the intervention per vaccine administered. This cost specifically refers to the intervention, and does not include the cost of the vaccine itself. Studies that did not unequivocally state the cost of the intervention per vaccinated person were not included in our cost-effectiveness comparisons. We have updated **Figure 7** to display the cost per vaccine.

*.... I keep thinking that the authors are mis-interpreting the main point of their paper: which is that bringing vaccines to villages for free and getting support from village authorities and having large community meetings and sending people to walk the village to talk with people (knocking on doors or holding small group meetings) all combined to dramatically increase vaccinations in these rural areas. A *bundle intervention* that included a lot of social support and information had a large effect on people who face large logistical (and perhaps social and informational) barriers to get vaccines like the COVID vaccine. Table 2 helps us see how this bundle of interventions compares to others but (1) the costs of the interventions in table 2 are largely costs in the urban and connected Global North and the costs of doing the same interventions in the rural Global South would be quite different, and (2) the costs and logistics of the village visit style intervention in urban Global North would differ from those described by this paper in Sierra Leone (I would love to see such interventions. I kept wondering what the equivalent approach in suburban USA would look like and whether if we could find the right authorities and mobilizers it would make a big difference or not).*

Thank you. We see your point, and in our revision, we focus more on simply and accurately describing what we did rather than adding our own interpretation that is not supported by the research design. By removing all the interpretive comments from the introduction, the reader's attention is now directed to Section 2 (page 3 and Figure 1) that simply states all the implementation steps that were taken, in chronological order. Those steps include a description of both the social mobilization, as well as our approach to addressing access barriers by taking vaccines and nurses to the remote villages.

On the call for replication: I would have liked to see a sentence or two about how those replications should differ so as to answer questions raised in this current study: is it really important to disentangle the bundle? I'm not sure it is, from the perspective of saving lives, but perhaps there are aspects of other interventions that might teach us more about how to encourage health behavior in relatively isolated places and/or more broadly.

Thanks for pointing this out. The revised manuscript is more precise about what the replications should be. Please see the revised **section “4.3. Policy Implications”**.

As we discuss towards the bottom of **page 1** of our Introduction, the same strategy can be applied to deliver bundles of maternal and child health interventions, including routine immunizations in those same communities. (Note that we are referring to something different with the word “bundle” than what you are referring to when you say “bundle”).

In fact, bundling multiple interventions would make the effort even more cost-effective than in our randomized controlled trial with COVID vaccines. That is precisely the next step of our program, leveraging the coalition we have formed through this project. We are now preparing for the implementation of this “bundled approach” to routine immunization with the Sierra Leone Ministry of Health and Sanitation.

All of this is important because routine immunization rates in Sierra Leone and neighboring countries in West Africa are low, and infant and child mortality rates too high, even relative to, say, Bangladesh. So our study results have broader applicability, and a potential to trigger large-scale action. In fact, our government partner is eager to implement this approach for the roll out of a new malaria vaccine.

F. Suggested improvements: experiments, data for possible revision

I recommend a re-framing of the paper. This is just re-writing. I would stop making the "demand side" versus "supply side" comparison since the intervention is not really clearly all "supply side" and since the research design does not actually randomize pure provision of vaccine versus the informational, social, aspects of vaccine take up and acceptance. That is, if there is a debate about supply versus demand, this paper is not well situated to address it. If local governments in the USA debate door to door vaccine provision versus SMS messaging, this is only in the context of bringing vaccines that much closer to people, and, in fact, I suspect that US governments are working on both at the same time --- both bringing vaccines as close to people as possible given infra-structure and personnel and trying to increase demand for vaccines. More importantly, I think that debate is a distraction from the goal of bringing vaccines and health to rural villagers in Africa. And I'm 100% compelled that the difficulty

of getting a vaccine deters most of these villagers from getting the vaccine.

Your message came through loud and clear, and we agree with it. We followed your advice very closely in re-framing the paper. The main value of our paper is indeed in demonstrating that (a) access difficulties are paramount in remote, rural areas of Africa, and (b) establish a proof of concept that it is indeed possible to overcome such access difficulties cost-effectively with the right coalition of partners and intervention design. That demonstration under real world conditions is valuable from both an academic and policy standpoint, because vaccination rates in rural African communities would not be in single digits even 2 years after covid vaccines arrived on the market if this was all straightforward and academics and policymakers believed that already. The concept we demonstrate via this RCT can be applied more broadly to deliver other important maternal and child health interventions including routine immunizations in remote, rural areas. We now make that case in a revised introduction and Section 4.3 "Policy Implications".

And I would have liked to see some of my confusions about the 45 villages and other small statistical questions cleared up.

Please see our response to your comment above.

G. References: appropriate credit to previous work?

Two more studies for Table 2. Both show small effects. "The effect of postcard reminders on vaccinations among the elderly: a block-randomized experiment." Chen, Trump, Hall, Le, Behavioral Public Policy, 2020. "Lessons for Covid-19 Vaccination from Eight Federal Government Direct Communication Evaluations" (a meta-analysis of 8 nudge style vaccine interventions) Kappes et al, Behavioral Science and Policy, 2021.

Many thanks for these suggested papers, they are included in the updated literature review, see **Table A10**.

H. Clarity and context: lucidity of abstract/summary, appropriateness of abstract, introduction and conclusions

I addressed this above. I think that this paper can be a powerful report on an important research design if the authors re-frame it. I'm eager to have this paper in the hands of governments, NGOs and anyone else active in vaccinations and public health.

Many thanks for your helpful comments. It is very motivating for our research team to receive your positive endorsement.

Response Memo: Reviewer 2

This paper reports the results of a cluster RCT mobile COVID-19 vaccination clinics in rural districts in Sierra Leone. The trial included a nested study in which some households within one of the intervention groups were randomized to home visits to provide vaccine information and encourage them to get vaccinated. The main findings that stand out in the paper are that the mobile clinics increased vaccination coverage in a community by about 20 percentage points (in the control group, vaccination coverage was ~5%) and that the home visits did not affect vaccination uptake.

Although the study provides useful information on what happens when mobile vaccination services are offered in places where there was little to no service delivery previously, I have several concerns regarding the novelty of the findings as well as the methods & reporting of results.

Thank you for your very constructive and helpful comments. We have taken your main comment about the framing of the paper seriously, and used it to re-frame our paper and entirely rewrite the introduction. It has helped us clarify the contribution. We discuss details under each of your points.

With regard to the novelty of this work – there are three points that could be taken into account in the framing of the paper:

One, it is well established and widely accepted that community-based delivery of health services in general can help increase access and uptake. The paper makes several statements in the Introduction and Discussion suggesting that the results from their study are novel in ways that should be surprising to researchers & policymakers seeking to increase vaccination rates in low-income countries. In fact, “reaching people where they are” or “bringing services closer to people” are not only commonly stated recommendations in health promotion efforts but also, they are already being practiced in many countries where there is adequate supply of vaccine doses. For high-income countries like the US, see for example the CDC guidelines for Health Departments on ways to increase COVID-19 vaccinations.

<https://www.cdc.gov/vaccines/covid-19/health-departments/generate-vaccinations.html>

More relevant to this study, see also the WHO Africa Region’s Monthly Bulletin (July 2022) on COVID-19 vaccination, which documents examples of many countries in which mobile clinics or community engagement approaches are being used to increase COVID-19 vaccination coverage. This could be noted in the paper and used to reframe the work as a trial that seeks to determine the effectiveness of such mobile service delivery approaches.

<https://apps.who.int/iris/bitstream/handle/10665/361671/CV-20220807-eng.pdf?sequence=1>

We now acknowledge in our revised introduction that WHO and CDC guidelines highlight the importance of “bringing services closer to the people”. Our paper provides rigorous scientific evidence that complements such guidelines. Our RCT is a proof of concept that such approaches can increase vaccination rates quickly and cost-effectively even under difficult circumstances in the most remote communities. Establishing that proof of concept is important because increasing vaccination rates in reality is much more complex than issuing guidelines about community outreach. It requires putting together a complex government-NGO coalition to creatively overcome logistical barriers. Our paper shows that it is possible to get that done and raise the number of people vaccinated by over 800% in 2-3 days. It demonstrates that result rigorously by setting up an RCT with a control group, and the paper further shows that such an effort can be cost-effective.

The fact that almost nobody is vaccinated in these remote communities 2 years after the vaccines is a relevant feature of the environment that requires global attention. That makes our demonstration and “proof of concept” necessary and useful. It is also a general feature not unique to the few communities where we work, in that millions of people across West Africa live in such remote communities.

We have now entirely rewritten the introduction to reframe along these lines and make these specific points. The revised introduction is a lot shorter and we have eliminated the statements that may indicate claims to conceptual novelty. We discuss in a bit more detail why this approach and proof-of-concept is important and different, in response to your comment #13 about another working paper that has implemented a related approach in Switzerland.

Two, the cleanest results in the paper are those for the outcome of vaccination rate (# vaccinated as percent of those enumerated in census). The finding here that about 20-25% of people get vaccinated comes as a surprise – instead of general tone of the paper suggesting that this was a major success, it actually comes as a surprise that mobile service delivery wasn’t even more successful. In many settings, one could argue that when vaccines became easily available – i.e. at nearby clinics or at mobile venues – uptake rose very rapidly to comparable percentages of 20-40%. There are various studies of mobile campaigns of other health services which have achieved extremely high uptake in rural communities – see for example the community-based testing strategies that are now widely implemented in many countries in Africa.

<https://www.nature.com/articles/nature16044>

To get a rigorous answer to your question about whether the increase in vaccination we observe is small or large, we have now conducted a much more thorough systematic review of all vaccination approaches published in the literature, as detailed in table A10 in our supplementary materials. This was a systematic review of articles in PubMed published between Jan 1, 2000 and Jan 7, 2023 using the search terms “(vaccin*[Title/Abstract] OR immun*[Title/Abstract]) AND additional search term [Title/Abstract] AND (Randomized Controlled Trial[Publication Type])”, with additional search terms: “access”; “community-based”; “cost effect*”; “demand”;

“hesitant”; “incentive*”; “intervention*”; “mobile”; “nudge*”; “rural”; and “supply”. These searches yielded 3,615 unique articles. After screening for articles that conduct an RCT, we identified 144 different published RCT studies that report the results of 234 unique interventions.

We found 3 access-focused vaccine studies that we label as "Community Actions" in our **Supplementary Table A10**:

- Sengupta et al. (2017) worked with "government Medical Officers responsible for [study areas] as well as the community...to establish a government funded outreach vaccination programme in settings provided by the community". Effect is 17.3pp for children fully immunised against seven VPDs after 1 year of the treatment, relative to pure control clusters.
- Ateudjieu et al. (2022): use monthly data recorded by community volunteers in the Fouban region of Cameroon to coordinate visits from a health center immunization team for catch-up immunization sessions. Effect is 0 (though it is worth noting the authors do not have a “pure control” like the one in our study, rather they compare two different approaches to community engagement).
- Habib et al. (2017): A community-based three-arm cluster randomised trial in three districts of Pakistan at high risk of polio. The two treatment interventions both included community outreach and mobilisation, using an enhanced communication package and provision of short-term preventive maternal and child health services and routine immunisation. Treatment effect, relative to a control group receiving routine polio programme activities is 7pp for the first treatment group, and 9pp for the second (which contains all the treatments bundled in treatment group 1, as well as the option to take the oral polio vaccine)..

While it may seem logical that mobile vaccine delivery should immediately reach the majority of the population, the fact that this was insufficiently prioritised by policymakers and commentators during the COVID-19 pandemic speaks to the necessity of further research on this topic. We believe that our study brings attention to and expands upon an underappreciated aspect of healthcare provision, which - as shown by our literature review - is not strictly academically novel, but we hope makes a forceful and rigorous argument for supply-side interventions as part of vaccinating remote populations. Your comment has led us to edit our manuscript in two further ways:

1. We cite the Nature 2015 Systematic review you mention on page 2 of the introduction to acknowledge the evidence on mobile delivery for HIV testing. We clarify why it is still important to establish the proof-of-concept for vaccination.
2. We have eliminated statements that may indicate claims to conceptual novelty
3. One implication of the different observed experiences with HIV testing versus COVID and other vaccinations is that underlying demand conditions may be very important for the success of access interventions. For example, if perceived mortality risk is high, people may be eager for the service, and an access intervention immediately produces large effects. We have therefore rewritten our entire introduction to move away the “demand vs supply framing”, which was a big theme in the previous version of our manuscript.

Third, it does not seem reasonable to compare across vaccine promotion studies in the way that the authors do in Table 2. Surely we need to recognize that demand side interventions may be necessary (and will have smaller effects) after the first wave of vaccine adoption takes place. I.e. supply side interventions like mobile service delivery in places where there was no supply previously will pick up the initial 20-40% who are interested in getting vaccinated, and after that one may need other interventions that target demand-side factors. The comparison of the mobile service delivery approach to other vaccination promotion interventions strikes me as apples and oranges – very different settings with very different baseline vaccination rates. For example, in the Campos-Mercade et al. RCT of incentives and behavioral nudges, baseline vaccination rates were 71%! How does it make sense to compare effectiveness of the incentives & nudges in that setting to the effectiveness of a mobile clinic in communities where barely anybody had been vaccinated initially? I would suggest paring back this section quite a bit and focusing on the results of your cluster RCT without too much additional commentary or comparison across studies. That could be a separate paper perhaps.

We are now more rigorous and systematic in how we compare our results to other vaccination efforts (see **Section 4.1 and Figure 6**) and in how we compare our costs to other studies that report cost-effectiveness numbers (see **section 4.2 and Figure 7**). We now explicitly acknowledge and highlight the point you make about the “apples to oranges” comparison in our introduction. We write,

“To benchmark our results against other vaccination strategies, we conduct a comprehensive systematic review that identifies 234 unique interventions in 144 RCT studies that use information, nudges, community engagement, social signaling, non-financial and financial incentives to increase vaccination rates. Over a third of these interventions produce null effects. Our access intervention produces a larger percentage point effect size than 218 (93%) of the treatments reviewed. This is not surprising, since vaccinating the first 50% of the population in remote parts of low-income countries requires solving the fundamental problem of access, which we address. Once access issues are addressed, misinformation and hesitancy may loom large in the effort to vaccinate the last 20% of the population of high-income countries who stubbornly hold out, and this is the target of the bulk of the literature.”

Major comments

1. *It seems odd for a primary outcome to be measured at baseline in the control group and at endline in the intervention groups. In a rigorous RCT, one would have a follow-up data from the control group as well. I recognize you are using baseline data since only 5-6 days elapsed, but still...it would almost have been better to collect data in control clusters at endline rather than baseline in this case. I think the comparison to baseline data from the control group is problematic.*

Thank you for your comment, which is well taken and we agree that ideally we would capture two measurements (baseline and endline) in each village. Our team considered the trade off, and the decision for one measurement was largely informed by the following logic. It is highly unlikely that these remote places visited by our vaccination teams would have been visited by other health personnel from MoHS in the ~5 day period between baseline and midline, or that a large number of people would have incurred the cost of visiting the CHC for receiving a vaccine. To substantiate this argument, we can use the fact that our baseline survey was conducted over a few weeks across communities, so we inspect the temporal trends in the data. We test the potential natural increase in vaccination rates over time by assessing the change in vaccination rates over the course of the rollout of the baseline survey. In the figure below we present the baseline vaccination rate (y-axis) over time including a regression line with 95% confidence intervals. There is no apparent trend, suggesting that vaccination rates did not increase over time and reducing worries that our choice of not revisit control villages impacts on the endline results.

We now acknowledge your concern and describe this test in the subsection **5.4 “Data Collection”** in our **“Methods”** section.

2. Rather than leading with Figure 2 in the results section, I would suggest presenting Figure 3 first since it is the primary outcome. I would also suggest reporting the control group vaccination rate in this figure

Thank you for this suggestion. We re-organized the presentation of the results. In **Section 3**, we now first present the *vaccination rate* pooled across treatment arms (**Figure 1**, ie for the village population as recorded in our census). In the next section we present the results for the *vaccination count* (**Figure 2**, ie for all those that were vaccinated regardless of residency). We now also present both the rate and count at baseline and endline rather than just the

Intent-To-Treat estimates, which are now presented in **Tables A2 and A3**. In **Section 3.3** we delve into the differences due to our demand side interventions and compare the rate and count for the village level Door to Door and Small Group campaigns (previously Table A2 and A3) and as well as the individual level Door to Door randomization, see Table 1.

The treatment effects in **Figure 2** and **Figure 3** are slightly larger than what we had estimated before. This is due to a small coding error and changes in the eligible population. The coding error related to a manual fix of the age variable in the data. We had used the exit survey of the vaccination drive to update respondent characteristics (predominantly age status). Some respondents were previously miscoded as being below 18 and thus not eligible for a vaccine. However, on inspection of the exit survey they were above 18. Because we did not have an equivalent additional check on age in control villages, this correction created a (downward) bias in our estimates. This has now been fixed resulting in slightly larger estimates.

We also followed R1's requests for additional information and test statistics. We now include an additional **Table A9** reporting on the difference between the full sample and the subsample for which we have survey data on attitudes and knowledge. Also, we have added multiple hypothesis testing q-values as well as bootstrapped p-values to our main **Tables A2, A3, A4 and A5**.

3. For measuring the primary outcome of vaccination uptake, should we be concerned about measurement error? I.e. the possibility that some individuals who were enumerated did get vaccinated but did not have their vaccination card with them?

Thank you for raising this point. Our primary outcome is the verified vaccine uptake measured using a respondent-level question on whether the person took a COVID-19 vaccine of any type and checked against their vaccination card. In total 4771 people received shots from the mobile teams, of these just 2 did not carry a vaccine card, so measurement error is very low. Measurement error at baseline may have been larger. At baseline 1,116 reported having had a COVID-19 vaccine for which we could verify the vaccine card. For an additional 1,555 (spread equally across treatment arms) we could not verify the vaccine card. These have been left out of the analysis (such as the count and rate values that appear in **Figure 2 and 3**). This implies we may be underestimating the baseline vaccination rate, but note this does not effect our estimate of the ITT. The verified vaccination rate is the standard measure of the Ministry of Health. Unverified vaccination rates could produce measurement error. Unverified vaccination rates could produce measurement error, as self-reports can be influenced by recall and social desirability bias.

4. Please report the power calculations and sample size determinations that were done at the outset of the study. This does not appear to be included in the paper.

Thank you for pointing out this important omission. In the revised version we added this to **Sub-Section 5.2** in the **Methods** Section. For our power calculation, we calculated the number of villages needed for a range of minimum detectable effect sizes assuming a 5% significance

level with 80% power. We expected decisions to take a vaccine should be highly correlated within a village, so we assumed an ICC of 0.15. In the 2015 census data, the mean population is 2480 people per village. Assuming that 50% of them were eligible to take the vaccine, we assumed an effective village size of 1200 people per village. Given that vaccine take up in Sierra Leone is low, we assumed a baseline vaccination rate of 2.5%.

Based on the treatment effects reported in the literature for similar studies and the large heterogeneity in treatment effects reported, we adopt a conservative approach and set our expected MDE at 0.05. We oversampled slightly and the final design included 150 communities across the three treatment arms, in 1:1:1 ratio. Given that we block-randomized within triplets, power naturally increases.

Minimum Detectable Effect	Number of Clusters Per Arm	# of Villages (control + experimental)
0.03	102	306
0.05	46	138
0.10	17	51
0.15	11	33

**We assume a constant effect size across all groups for the computations above.*

5. The effectiveness of the home visits would seem to depend quite a bit on implementation factors – i.e. fidelity to the intervention and time spent at each house that was selected – as well as the quality of the information & interactions. How was the curriculum developed? Did the team collect information on implementation factors?

In the two sub-treatments being compared - the “Door to Door” or “Small Group mobilization” the protocol, information content, and the messaging that Social mobilizers were trained to provide were exactly the same. The main difference between these two was the selection of target recipients. There were also minor differences in the way the teams were trained to introduce themselves.

In the “Small Group mobilization” treatment, our teams went to specific areas where they expected to encounter groups of people. Religious buildings like mosques during prayer times, eating places, water sources, farm huts, etc. are representative examples. They could learn more about where to find groups of people from the chief or a local elder, but they were also empowered to just approach groups of people anywhere that a group had already formed.

For the “Door to Door mobilization”, social mobilizers were assigned a set of specific housing structures to visit, where those structures were randomly chosen from our listing. and follow the same protocol.

Unfortunately, we did not collect systematic data on implementation factors like “effort” or “quality of interactions”. We set up close supervision of the field teams to try to ensure high quality implementation in both treatment arms. Field notes from our field coordinators notes no appreciable differences in the implementation effort or quality between the treatment arms. Of course, despite our observations and elaborate training, implementation factors could explain our null finding on the marginal effect of home visits, which we now acknowledge in **Section 3.2**.

6. Did the study measure distances from households to the vaccination site? If yes, it would be useful to report average distances and examine whether distance is associated with uptake. If no, then it would be helpful to give an indication of approximate travels times/costs associated with coming for a vaccine.

Villages in our study sample are small, and people generally had to walk very short distances from their homes to where the vaccine team had set up at a central location in the village. Distances within villages vary little. Once the access treatment was implemented in a village, the cost of getting from the home to the vaccination clinic in the village center is small. We now make this point in **Section 2 on page 3**.

7. Doesn't the possibility of spillover effects across households reduce the ability to detect an effect of the home visits (Table 1)? I would suggest discussing this possibility and also adjusting for it if possible (possible with spatial adjustment if there is information on proximity to treated households).

To learn about the effect of home visits we have two sources of information: our between and within-village randomization. The village level randomization allows us to learn about the effect of assigning whole villages to receive Door to Door visits by mobilization teams. These estimates should remain unaffected by spillover effects. Spillover between villages is likely very small because there's limited travel between treatment and control villages in these remote areas. During the exit survey, enumerators asked respondents who reported to have come from a different village about the District and Village name. We then matched these names back to our list of control villages. Using a hard match on District names and then a Levenshtein distance metric to match village names (this is a text similarity measure that compares two words and returns a numeric value representing the distance between them). Allowing for a string distance of 2, we find only 8 matches. Using a more conservative cut-off of 1, find no overlap whatsoever. In addition, the minimum as the crow flies distance between project villages was 8.5 miles, so at least a 2-3 hour walk. We now mention this in Section 5.4 on **page 22**.

Spillovers within villages are more likely in theory. We unfortunately lack data on distances and other channels of interactions between households to test this. We however believe the effect is likely small, as turnout to community meeting was already large (ie people on the margin were likely already convinced at the village meeting), social mobilizers were visiting people in their homes for a private conversation (vaccine decisions are a private concern, making it less likely that neighbors would then talk to each other about it), there was relatively short time between social mobilization activities and the vaccine drive (if people needed time to influence each other we would likely have missed this in our design). We now acknowledge this possibility of within-village spillovers in our manuscript in **Section 3.2 on page 9**.

“Within village spillovers may also dampen these individual treatment effects. We unfortunately lack data on distances and other channels of interactions between households to test this formally. However, this type of spillover may be small due to the relatively short time interval between the home visits and the vaccine drive.”

a. Minor point: the term “door knock” as used in the paper and in the tables is rather awkward and casual. Consider replacing it with terms like “Home visits” or “interpersonal communication”

Thanks and we agree. We have changed the language in the paper and now refer to the Door to Door campaign or as suggested Home Visits when referring to the outreach work of mobile teams.

8. Regarding the authors’ recommendation that community meetings be widely implemented, how practical is this and shouldn’t the CEA analysis adopt a household perspective and account for costs incurred by households as well any costs that would be incurred in order to increase meeting attendance?

There are two types of costs that the referee highlights:

1. The cost to the implementer of organizing these meetings.
2. Costs borne by households when they attend meetings.

The marginal cost of organizing the meetings was essentially zero, because our implementation team has to be present in the village for 48-72 hours anyway, and there is a lot of down-time during the period that they are in the village. We make use of their down-time to organize the meetings, and we think more of the down-time can be efficiently used to put more effort into encouraging village residents to attend meetings. The implementation staff are paid per-diems for the entire period that they spend in the village anyway, and as described in the manuscript, the lion’s share of implementation costs is the transport cost of getting to the village. Once the transportation cost is paid, it is sensible to use the team’s time in the village more effectively.

The referee is correct that asking village residents to attend meetings imposes on their time. These costs appear to be quite small in this context because the villages are small. In the typical village, houses are often clustered together. Generally speaking, most people have to walk less than a couple of hundred meters to attend the meetings. To try and minimize the inconvenience, we choose the time and place for the meetings in consultation with the community. This is why our community meetings are most often held in the early evenings after people returned from their farms. The opportunity cost of time is low during that specific time slot.

The referee is correct that depending on context, meetings can be inconvenient or costly to attend. We now acknowledge this point in the cost-effectiveness section on **page 15**.

“Note that here we are looking at cost effectiveness from the perspective of the planner (ie the government), and do not consider the costs of households. Depending on context, meeting attendance can be inconvenient or costly. In our context, villages are small. On average, people had to walk less than a couple of hundred meters to attend the meetings. Also, to minimize the inconvenience, meetings were held in the early evenings after people returned from their farms. As a result, the opportunity cost of time is low for most of our participants.”

9. What COVID-19 vaccine did the NGO offer in the communities? Important information that doesn't seem to be mentioned anywhere.

Thanks for pointing out this important omission. We now mention the COVID-19 vaccine type on **page 7, footnote 2**. In total the teams vaccinated 4771 people aged 12 or above. Of these 39% received a Johnson & Johnson vaccine, 29% Pfizer, 17% Sinofarm and 15% received AstraZeneca.

10. What was the search strategy used to identify “recent studies” that sought to increase vaccine uptake? This does not appear to be described in the paper and I think it is important to explain the databases and search terms that were used. Also, I suggest that Table 2 have a different title other than “comparing vaccination rates”.

We have updated our literature overview and now present the results of a systematic review of RCTs that include approaches to increase vaccination uptake. We included a new **subsection 5.7** in Methods section that describes our search procedure.

We searched PubMed of papers published between Jan 1, 2000 and Jan 7, 2023 using the search terms "(vaccin*[Title/Abstract] OR immun*[Title/Abstract]) AND \textit{additional search term}[Title/Abstract]) AND (Randomized Controlled Trial[Publication Type])", with additional search terms: ``access"; ``community-based"; ``cost effect*"; ``demand"; ``hesitant"; ``incentive*"; ``intervention*"; ``mobile"; ``nudge*"; ``rural"; and ``supply". These searches yielded 3,615 unique articles. We screened out articles that are not related to vaccine uptake or that did not rely on an RCT, reducing the sample to 141 articles. We appended a further 20

relevant studies that were identified by snowballing and rejected 17 papers that did not have a control group; did not report the percentage-point change in vaccine uptake; and did not include a test statistic. The final list of 144 articles comprise 234 unique interventions for which we can report a percentage-point change relative to a control group.

We then grouped these studies as information, nudges, community engagement, social signaling, non-financial and financial incentives. The mean effect sizes per group is included in the new **Figure 6**. The full list of papers (an expanded version of what used to be Table 2 in the old version of the manuscript) is now in the Supplementary Materials, as **Table A10** with the more appropriate title (responding to your comment above): “Effect Size of Previous Vaccine Uptake Studies”

Of the final list, 33 interventions (14%) report information about the cost of the intervention per vaccine administered. This cost specifically refers to the intervention, and does not include the cost of the vaccine itself. Studies that did not unequivocally state the cost of the intervention per vaccinated person were not included in our cost-effectiveness comparisons. We have updated **Figure 7** to display the cost per vaccine.

11. Were the sub-group analyses pre-specified? I'd recommend indicating this in the methods including the exact sub-groups that were pre-specified.

We had not pre-specified the subgroup analysis we include in **Figure 5**. We included demographic subgroups often used in vaccination studies (age, education and gender) and some income proxies (land ownership and food security). The logic is that given the high cost of accessing vaccines without our intervention, wealth status is likely to be a mediating factor. We now have a **Section 5.8** in our Methods section titled “Deviations from Pre-registered hypotheses” where we explicitly note that these heterogeneity tests were not pre-specified, see **page 26**.

“The heterogeneity analysis reported in Figure 5 where we study whether vaccination rates differ by age, gender, schooling, and wealth, was not pre-specified. These heterogeneity tests are common in the vaccine literature, see Lazarus (2021)”

12. Community and home-based service delivery have been compared to facility-based service delivery in many contexts and it is widely accepted that bringing services closer to people increases access and uptake. Consider citing other studies that have shown such results. Here is a review article in Nature that establishes this for HIV services <https://www.nature.com/articles/nature16044>

Thank you. We have now cited the HIV article in our introduction on page 2. We have also acknowledged your point in the introduction by adding the text below, but we explain why it is important to generate rigorous evidence on vaccination in remote, rural settings, and establish a

proof-of-concept that vaccinations can be delivered even to remote places at reasonable, and that solving the access issue raises vaccination rates quickly.

“CDC and WHO guidelines highlight the importance of “bringing services closer to the people”, and our RCT is a proof of concept that such approaches can increase vaccination rates quickly and cost-effectively even under difficult circumstances in the most remote communities. The mobile delivery concept has produced large effects on HIV testing (Sharma et al., 2015), but rigorously demonstrating effectiveness in vaccine delivery is critical, given the persistent low rates of vaccination in low-income countries.”

13. Cite other studies have done similar comparisons of mobile vs clinic based vaccination – e.g. this one from Switzerland:

https://papers.ssrn.com/sol3/papers.cfm?abstract_id=4166827

We describe the inclusion criteria for our systematic review above, in response to your comment #10. Note that one of these criteria is “studies published between Jan 1, 2000 and Jan 7, 2023, which is why this study does not appear in the long list of studies included in **Supplementary Table A10**. Nevertheless, we have now cited this paper in our introduction. Hopefully our systematic review has captured all other relevant citations.

Beyond citing the Swiss paper, we should also engage with the substance of your comment, which is linked to other comments that appear elsewhere in your report. As we understand it, your concern is generally about novelty:

“other studies have shown that...” or “it is widely accepted that...” “mobile service delivery that brings services closer to the people increases uptake.”

You are correct that our study and the Swiss study have one thing in common - they both demonstrate that access is important. The Swiss study demonstrates that making access convenient in a high-income country in the earliest phase of vaccine distribution (August 2021 in Switzerland) speeds up uptake. But there are some fundamental ways in which our study, its purpose, and its context differs from this Swiss working paper. Unlike Switzerland and other high-income countries, where everyone who wanted a vaccine could easily access one by late 2021, the vaccine still has not reached the majority of the African population. The logistical challenge of last-mile delivery in remote rural parts of Sierra Leone is fundamentally different from sending a van down paved roads in Switzerland.

It is important to demonstrate that this challenging task can also be completed cost-effectively in remote parts of Sierra Leone for reasons beyond the academic goal of establishing rigorously that access is important. Establishing that proof-of-concept in the most remote logistically challenging context allows us to mobilize partners and build the necessary coalitions of donors, health ministries, NGO partners, and researchers to replicate and scale that same implementation strategy in other remote regions of Sierra Leone and in other low-income

countries. This is also why we do the additional work of benchmarking or effect size and cost-effectiveness against other vaccination efforts around the world, to make the case that this can actually be done at scale cost-effectively, and we will reap the returns.

And the potential applications of this proof of concept go beyond COVID vaccines. As we discuss towards the top of page 2 of our Introduction, the same strategy can be applied to deliver bundles of maternal and child health interventions, including routine immunizations in those same communities. In fact, bundling multiple interventions would make the effort even more cost-effective than in our randomized controlled trial with COVID vaccines. That is precisely the next step of our program, leveraging the coalition we have formed through this project.

All of this is important because routine immunization rates in Sierra Leone and neighboring countries in West Africa are low, and infant and child mortality rates too high, even relative to, say, Bangladesh. So the broader applicability of our study, to trigger large-scale action is fundamentally different from a study conducted in Switzerland. And demonstrating in the real world that it can be done cost-effectively serves a fundamentally different purpose than writing down in a WHO or CDC guideline document that this is conceptually a good idea to try.

14. I am surprised that aside from the director of the EPI program in Sierra Leone, there were no local academic researchers or senior research staff involved as co-authors.

Thank you for raising the concern. We appreciate your thoughtful feedback with respect to authorship. To provide more background, the Director of the EPI program in Sierra Leone is also an Associate Lecturer at the department of Environmental and Health Sciences within the School of Public Health at Njala University in Sierra Leone, so he participated in this study not only as an implementer, but also as an academic researcher.

We wholeheartedly agree with you that recognition of the effort of our wider team is of utmost importance. We have added our two senior field research staff as co-authors, who previously were named in our “Acknowledgement” section. Our oversight comes from the history in social science publication culture, where norms of authorship are different. The two added co-authors played a central role in data acquisition and team management, and meet the *Nature* guidelines for authorship.

15. In line with my earlier points, abstract could revised to report the main methods & results. These sentences seem better suited for discussion/interpretation – the second sentence doesn't seem factually true given that the study did not achieve anywhere close to universal coverage. “This has led to an explosion of scholarly and policy interest on vaccine hesitancy and ways to overcome it. We argue instead that the limiting factor for many rural Africans is lack of access to convenient vaccine services, and implemented an intervention designed to overcome the logistical challenges of last mile service delivery”

We have rewritten the abstract and we have eliminated this language.

Minor comments

1. *It would be useful to report the exact % vaccinated in each to the three clusters in addition to the regression results. Currently it is not easy to know the exact % of population vaccinated in each of the two intervention groups (for the control group you report the % in the regression results in Fig 2 and one has to use the regression estimates in Fig 3 to work this out). I would recommend replacing Figure 3 with a simple table that shows vaccination rates in each study group (including control) and the effect size of the interventions as betas.*

Thank you for these suggestions. In the Figures we now include the baseline rate in **Figure 2** and count in **Figure 3**. And in **Supplementary Tables A2 and A3** report the regression output underlying the figures. In the new Table 1 we report the estimates for each treatment arm separately, as well as the effect of the within village Door to Door randomization.

2. *There are several places with text that is unnecessary or somewhat removed from the main focus of the paper. For example, there is a discussion on page 18 about whether vaccination is necessary. This strikes me as besides the main goal of this paper, which is a scientific study of whether the mobile clinic strategy increases vaccination coverage and the corresponding cost-effectiveness of that strategy. I suggest reducing the length of this already long paper by focusing on the main analytical components of the paper.*

We have re-written **section 4.3**, Policy Implications, and we have footnoted this point. **Section 4.3** is now better focused on the main goal of the paper.

3. *There is repetition of text in various places. Reducing such repetition would make the paper much easier to read. Examples include:*

a. *Pubmed searches about demand side vs supply side approaches (repeated in Introduction and Discussion)*

b. *In the Results section, the vaccination rate is defined multiple times – with the point about the census & denominator being mentioned on page 6 and 7. I would suggest clearly defining the vaccination rate in Section 2 and then reporting the results only in the Results section.*

Thanks for pointing these out. We have streamlined the text. The “demand versus supply” framing and discussion has been eliminated from the introduction.

Our outcome variables, ie the vaccination rate and count are now more clearly defined at the top of Section 3.

4. *Reduce the number of digits in the tables, especially those in the Appendix*

In the revised version we have made all tables consistent with *Nature* formatting guidelines. We did not see a specific guide as to the number of digits, but checked several published papers and have consistently restricted decimals to the thousands.

5. Remove statistical tests of differences between study groups from Table 1. These don't seem necessary to me given that this was an RCT.

Thank you for pointing this out. We have removed the variable level t-tests from the descriptive statistics and balance table.

6. Consider using past tense when describing the study results. Currently there is a mix of present and past tense being used (with present tense more commonly used) – uniformity in the language would make the paper easier to read.

Thanks for pointing this out. In the revised version we now consistently use past tense when referring to our results.

7. Consider using "increased to" instead of terms like "vaccination rate jumps to.." Our quantity of interest

Thank you for flagging this. We have changed the language in the paper according to your suggestion.

Response Memo: Reviewer 3

This paper offers a compelling argument to dispel the vaccine hesitancy myth drawing on the results of the Expanded Programme on Immunization in Sierra Leone. The 20% increase in vaccine up take supports the primary argument presented in the paper demonstrating that people living in rural Sierra Leone were not deeply hesitant and that providing access to vaccines was an effective way to increase vaccination coverage.

The article's reflections on academic literature and funding priorities that reinforce the perception of vaccine hesitancy across the African continent, is profoundly important in attempting to reshape the narrative on vaccine access. The partnership with the Sierra Leone Ministry of Health, further emphasizes the critical role of working alongside governments to ensure access to vaccines and healthcare more generally.

Very importantly, the article demonstrates the value of working with community leaders in rural areas to ensure vaccine uptake, reinforcing the need to plan interventions with communities.

Based on this assessment, I recommend the paper for publication.

Thank you for your positive assessment of the paper. In light of the comments from the other referees we have revised the paper. Given your clear interest in the policy actions that governments and the international community should take that are related to our research results, we have now reframed the introduction and the “policy implications” section (4.3) accordingly. In that section we now write:

The most immediate and direct implication of our results is for the government of Sierra Leone to replicate and expand this cost-effective program to reach the 59% of the country's population who reside in similar remote, rural areas outside of PHU coverage. The bulk of our intervention cost is the transport cost of reaching remote communities, so an obvious implication is that we should bundle Covid vaccines with other necessary mother/infant/child health interventions that can be delivered simultaneously on the same trip. That could dramatically reduce costs per person treated. This would still be expensive for a resource-constrained MoHS to launch at scale, and international partners must provide support.

We have begun building the necessary coalition to implement similar strategies to deliver a bundle of vaccines to improve the cost-effectiveness and scalability of this last-mile-delivery intervention. Sierra Leone MoHS has prioritized HPV vaccination for girls aged 10-12, and routine immunizations (DTP, Measles, Polio) for children aged zero to six to bundle with any further COVID-19 vaccine delivery. Operationally, this leverages existing (but underutilized) clinic infrastructure in a hub-and-spoke model to provide mobile vaccination services

near citizens' doorsteps, and bring health services more cost-effectively to the most remote communities that currently lack access.

END OF RESPONSE LETTER

Reviewer Reports on the First Revision:

Referees' comments:

Referee #1 (Remarks to the Author):

Because this is a review of a revision, I am not going to re-iterate what I said before. The authors of this article responded to all of my concerns beautifully and in great detail. I am impressed at how they used the comments from the reviewers and editor to really improve their paper.

Referee #2 (Remarks to the Author):

I appreciate the authors' carefully thought out responses to my comments and the many revisions to the paper. Overall this is a much stronger version and a compelling paper to read. I have a few other comments for the authors to consider:

1. The paper may need to more clearly present the 'treatment' (perhaps in the abstract if not the title, and elsewhere) as a multi-component intervention that is not only about bringing vaccines to villages. The community mobilization component may need equal emphasis and the discussion of mechanisms suggests that this may well be an important contributor to the overall treatment effect. In other words, do you want readers to take away that simply showing up and making vaccines available is sufficient? Or do you want to tell readers that you need to mobilize the community (with meetings) and then deliver the vaccines?
2. Were the social mobilizers employed by the research team or were they government (i.e. MOH) employees? If it was the former, it is worth noting in the discussion that one of the limitations is that we need to test such approaches in settings where gov't officials/staff run such meetings rather than researcher-led teams.
3. The revised introduction does a better job of explaining the rationale for the study and situating its contributions to the literature and to global health policy.
4. I appreciate the authors' detailed response to my comment about the novelty/significance of this work. I should clarify that in citing the Switzerland study I was not making the point that we should look to that study for all the guidance we need. Instead, my larger point remains that the results of this study will not surprise many people working in global health because it is well known that mobile service delivery is a far better alternative to fixed facility delivery. Going back to the WHO Bulletin from July 2022, there is a clear discussion of many countries turning to mobile vaccination strategies to get the vaccines out to people. So my question is that setting aside any concerns about novelty, is the reluctance to do more mobile service delivery stemming from not knowing about 20-30% effect sizes, or is it because many African countries unfortunately did not have sufficient doses to prioritize mobile service delivery?

5. On page 15 when discussing Table A10, it would be worth noting that very few studies have taken place in LMICs. The vast majority are in high-income countries.

6. I don't find it very useful to compare cost-effectiveness data from US/European demand-side interventions for COVID-19 vaccination. I'd suggest paring back this text.

7. Your point about the value of combining mobile COVID-19 vaccine delivery with the provision of other essential health services is well-taken. There is in fact some ongoing work on multi-disease mobile health campaigns or integrated health campaigns in Africa – e.g., see <https://journals.plos.org/plosone/article?id=10.1371/journal.pone.0012435>.

8. I'm surprised there were so many different vaccines used despite the study's implementation over a 2-month period. What was the rationale for this?

Author Rebuttals to First Revision:

Nature MS #2022-09-14388A

“Solving Last-Mile Delivery Challenges is Critical to Increase COVID-19 Vaccine Uptake: A Cluster Randomized Controlled Trial”

Response to reviewers

Referee #1 (Remarks to the Author)

Because this is a review of a revision, I am not going to re-iterate what I said before. The authors of this article responded to all of my concerns beautifully and in great detail. I am impressed at how they used the comments from the reviewers and editor to really improve their paper.

Thank you for your thoughtful comments, which greatly improved the paper.

Referee #2 (Remarks to the Author)

I appreciate the authors' carefully thought out responses to my comments and the many revisions to the paper. Overall this is a much stronger version and a compelling paper to read. I have a few other comments for the authors to consider:

Thank you for your thoughtful comments, which greatly improved the paper.

1. The paper may need to more clearly present the ‘treatment’ (perhaps in the abstract if not the title, and elsewhere) as a multi-component intervention that is not only about bringing vaccines to villages. The community mobilization component may need equal emphasis and the discussion of mechanisms suggests that this may well be an important contributor to the overall treatment effect. In other words, do you want readers to take away that simply showing up and making vaccines available is sufficient? Or do you want to tell readers that you need to mobilize the community (with meetings) and then deliver the vaccines?

Thank you. This is a fair point. We have revised the abstract, and added text to the introduction to explicitly acknowledge that this is a multi-component intervention that includes community mobilization. We also describe the mobilization activities in detail in Methods subsection “Intervention details”, on page 21 of the PDF. The new text added to abstract and intro is highlighted in red:

New Abstract Text:

...Motivated by the observation that residents of remote, rural areas of Sierra Leone faced severe access difficulties, we **conducted an intervention with last-mile delivery of doses and health professionals to the most inaccessible areas, along with community mobilization**. A cluster randomized controlled trial in 150 communities shows that **this intervention with mobile vaccination teams** increases the vaccination rate by about 26 percentage points within just 48-72 hours...

New Text in Introduction:

"The centerpiece of this intervention was to simply take vaccine doses and nurses to administer vaccines to remote, rural communities, **preceded by some community mobilization activities**."

The editor gave us some specific guidance for the title, which needed to be shorted to less than 75 characters. So we do not have sufficient space to add "community mobilization" to the paper title. The new suggested title now reads "Last-Mile Delivery Increases Vaccine Uptake in Sierra Leone".

2. Were the social mobilizers employed by the research team or were they government (i.e. MOH) employees? If it was the former, it is worth noting in the discussion that one of the limitations is that we need to test such approaches in settings where gov't officials/staff run such meetings rather than researcher-led teams.

We added the following text early in the "Context" section to explain who these community mobilizers are:

"The social mobilizers were previously vetted and trained by ministry staff, and commonly engaged for short-term projects like vaccination campaigns. This cadre is referred to as "MoHS volunteers" because they are paid per-diems against project work, and not a regular civil servant salary."

In the Discussion section, where we discuss scaling, we have added the following caveat:

"The wide availability of a cadre of staff known as "Ministry of Health volunteers" - individuals already vetted by the ministry but not on their regular payroll and available to work as mobilizers on special projects against per-diems -- increases the potential for scaling this project nationwide in Sierra Leone. One potential challenge of replicating this project to other countries is to find substitute staff who can take on that mobilization role."

3. *The revised introduction does a better job of explaining the rationale for the study and situating its contributions to the literature and to global health policy.*

Thank you.

4. I appreciate the authors' detailed response to my comment about the novelty/significance of this work. I should clarify that in citing the Switzerland study I was not making the point that we should look to that study for all the guidance we need. Instead, my larger point remains that the results of this study will not surprise many people working in global health because it is well known that mobile service delivery is a far better alternative to fixed facility delivery. Going back to the WHO Bulletin from July 2022, there is a clear discussion of many countries turning to mobile vaccination strategies to get the vaccines out to people. So my question is that setting aside any concerns about novelty, is the reluctance to do more mobile service delivery stemming from not knowing about 20-30% effect sizes, or is it because many African countries unfortunately did not have sufficient doses to prioritize mobile service delivery?

You raise a very interesting question – if the effect sizes are indeed large, and if this is a cost-effective activity, then why aren't more countries implementing the strategy? Why have vaccination rates remained so low - especially in remote rural areas of Africa?

Our experimental design does not permit a rigorous answer to this question, but in the manuscript we hypothesize - based on our fieldwork experience - that Ministries of Health need to experiment with novel strategies and engage in learning by doing. Experimentation requires resources, and making more doses and resources available to developing country governments would have helped. This is the text we added in the on policy implications:

“A broader implication for international development partners and pharmaceutical companies is that they need to facilitate and underwrite such experimentation by making vaccine doses and budgets readily available to allow ministries of health to learn what approaches work best in a given context. Local institutions need to engage in “learning by doing”, which is impossible without a reliable supply of vaccines, and incentives for staff to tinker with innovative ideas.”

5. *On page 15 when discussion Table A10, it would be worth noting that very few studies have taken place in LMICs. The vast majority are in high-income countries.*

Thank you. Note that Table A10 is now in the Supplementary Information section, we have amended this text:

“Supplementary Information Section 2 provides the details of the intervention approach used in each study. **The vast majority of these studies were conducted in high income settings.**”

6. I don't find it very useful to compare cost-effectiveness data from US/European demand-side interventions for COVID-19 vaccination. I'd suggest paring back this text.

We have deleted part of the text in this section. We only retained comments about one study that was conducted in rural India, which is a more comparable setting to ours. We also retained some (pared-back) text about the Campos-Mercade study, because that was the only other study about COVID-19 that provides cost information, so other readers may complain if we don't explicitly acknowledge it and cite it here.

~~Of particular interest is the cost-effectiveness of alternative Covid-19 vaccine strategies. The \cite{milkman_city_2022} Philadelphia lottery was particularly costly (US\$4,485 per person vaccinated) because \$63,000 worth of lottery prizes produced a very small increase in vaccinations.~~

A study in rural India (Banerjee et al 2010) pursues a similar strategy to ours by setting up measles vaccination clinics. That treatment costs US\$75 (in 2022 dollars) per vaccine administered but adding an incentive for the parents to bring their children to the clinic lowers the cost to US\$38 per child vaccinated. The only other COVID-19 vaccine study in our systematic literature review to provide cost information (Campos-Mercade et al 2021) offered US\$24 as a financial incentive to get vaccinated in Sweden. Unfortunately, this study does not report the costs of other program components, such as the cost of administering the incentive program, verifying individual-specific vaccination information in the administrative records, sending two text message reminders, etc.

~~Although long-standing vaccines such as those for influenza and HPV have received academic attention, the only other COVID-19 study identified within our systematic literature review to provide cost information is that of \cite{campos-mercade_monetary_2021}—which offered US\$24 as a financial incentive to get vaccinated in Sweden. This comes close to identifying the cost per vaccine administered, but stops short of explicitly specifying the actual figure. It omits possible costs of other program components, such as the cost of administering the incentive program, verifying individual-specific vaccination information in the administrative records, sending two text message reminders, etc.~~

~~Studies that did not clearly and unequivocally state the cost per successfully administered vaccine were excluded from the figure. This means that studies that only reported the overall cost of the project, or simply mentioned the size of the financial incentive offered, were~~

~~disregarded for the purposes of making this figure. This allowed us to avoid making inaccurate assumptions about the presence or absence of hidden (unreported) costs.~~

7. Your point about the value of combining mobile COVID-19 vaccine delivery with the provision of other essential health services is well-taken. There is in fact some ongoing work on multi-disease mobile health campaigns or integrated health campaigns in Africa – e.g., see <https://journals.plos.org/plosone/article?id=10.1371/journal.pone.0012435>.

Thank you for bringing this interesting paper to our attention. We now cite it in the Discussion section, where we discuss the “bundling” idea.

A recent study in rural Western Kenya (Lugada et al 2010) demonstrates that such integrated approaches combining HIV testing with other preventative health services like bednets and water filters can be successfully implemented.

8. I'm surprise there were so many different vaccines used despite the study's implementation over a 2-month period. What was the rationale for this?

We added the following sentence in section “Effects on Total Vaccination Count” to explain:

“Of these 39% received a Johnson & Johnson vaccine, 29% Pfizer, 17% Sinofarm and 16% received AstraZeneca. **A variety of vaccine types were administered because there was no steady supply of any specific type of vaccine dose in Sierra Leone when this intervention was conducted, so we had to make use of whatever product was available in the Ministry of Health stocks in any given week.**”

Our partners MoHS/EPI were receiving vaccines from different facilities and donations - e.g. CoVAX was mostly providing AstraZeneca and Pfizer, the Chinese Embassy was providing Sinovac, and the Government procured J&J with World Bank funds, and that explains the variation in vaccine types.